# Flexible Diffusion Modeling of Long Videos

**William Harvey, Saeid Naderiparizi, Vaden Masrani, Christian Weilbach, Frank Wood**[*]
Department of Computer Science
University of British Columbia
Vancouver, Canada
{wsgh,saeidnp,vadmas,weilbach,fwood}@cs.ubc.ca

## Abstract

We present a framework for video modeling based on denoising diffusion probabilistic models that produces long-duration video completions in a variety of realistic environments. We introduce a generative model that can at test-time sample any arbitrary subset of video frames conditioned on any other subset and present an architecture adapted for this purpose. Doing so allows us to efficiently compare and optimize a variety of schedules for the order in which frames in a long video are sampled and use selective sparse and long-range conditioning on previously sampled frames. We demonstrate improved video modeling over prior work on a number of datasets and sample temporally coherent videos over 25 minutes in length. We additionally release a new video modeling dataset and semantically meaningful metrics based on videos generated in the CARLA autonomous driving simulator.

## 1 Introduction

Generative modeling of photo-realistic videos is at the frontier of what is possible with deep learning on currently-available hardware. Although related work has demonstrated modeling of short photo-realistic videos (e.g. 30 frames [36], 48 frames [6] or 64 frames [16]), generating longer videos that are both coherent and photo-realistic remains an open challenge. A major difficulty is scaling: photorealistic image generative models [4, 8] are already close to the memory and processing limits of modern hardware. A long video is at very least a concatenation of many photorealistic frames, implying resource requirements, long-range coherence notwithstanding, that scale with frame count.

Attempting to model such long-range coherence makes the problem harder still, especially because in general every frame can have statistical dependencies on other frames arbitrarily far back in the video. Unfortunately fixed-lag autoregressive models impose unrealistic conditional independence assumptions (the next frame being independent of frames further back in time than the autoregressive lag is problematic for generating videos with long-range coherence). And while deep generative models based on recurrent neural networks (RNN) theoretically impose no such conditional independence assumptions, in practice they must be trained over short sequences [12, 26] or with truncated gradients [31]. Despite this, some RNN-based video generative models have demonstrated longer-range coherence, albeit without yet achieving convincing photorealistic video generation [26, 3, 7, 20, 2].

In this work we embrace the fact that finite architectures will always impose conditional independences. The question we ask is: given an explicit limit $K$ on the number of video frames we can jointly model, how can we best allocate these frames to generate a video of length $N > K$? One option is to use the previously-described autoregressive model but, if $K = N/4$, we could instead follow Ho et al. [16] by training two models: one which first samples every 4th frame in the video, and another which (in multiple stages) infills the remaining frames conditioned on those. To enable

---

[*]Frank Wood is also affiliated with the Montréal Institute for Learning Algorithms (Mila) and Inverted AI.

36th Conference on Neural Information Processing Systems (NeurIPS 2022).

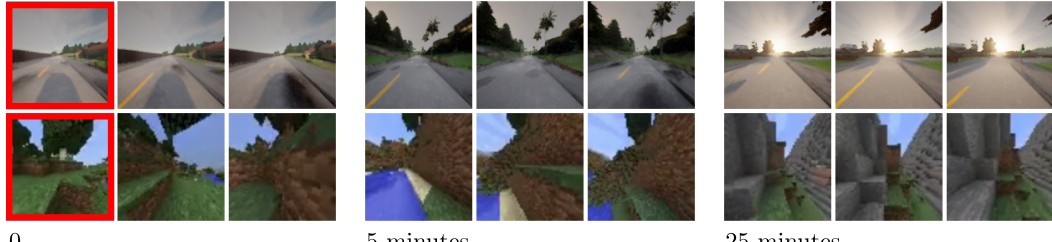

0                           5 minutes                      25 minutes

Figure 1: A long video (25 minutes, or approximately 15 000 frames) generated by FDM for each of CARLA Town01 and MineRL, conditioned on 500 and 250 prior frames respectively. We show blocks of frames from three points within each video, starting from the final observed frame on the left. Blocks are marked with the time elapsed since the last observation and frames within them are one second apart. We observe no degradation in sample quality even after > 15 000 frames.

efficient exploration of the space of such sampling schemes, we propose a flexible architecture based on the denoising diffusion probabilistic model (DDPM) framework. This can sample any subset of video frames conditioned on observed values of any other subset of video frames. It therefore lets us explore a wide variety of previously untested sampling schemes while being easily repurposed for different generation tasks such as unconditional generation, video completion, and generation of videos of different lengths. Since our model can be flexibly applied to sample any frames given any others we call it a Flexible Diffusion Model, or FDM.

**Contributions** **(1)** At the highest level, we claim to have concurrently developed one of the first denoising diffusion probabilistic model (DDPM)-based video generative models [16, 40]. To do so we augment a previously-used DDPM image architecture [15, 22] with a temporal attention mechanism including a novel relative (frame) position encoding network. **(2)** The principal contribution of this paper, regardless, is a "meta-learning" training objective that encourages learning of a video generative model that can (a) be flexibly conditioned on any number of frames (up to computational resource constraints) at any time in the past and future and (b) be flexibly marginalized (to achieve this within computational resource constraints). **(3)** We demonstrate that our model can be used to efficiently explore the space of resource constrained video generation schemes, leading to improvements over prior work on several long-range video modeling tasks. **(4)** Finally, we release a new autonomous driving video dataset along with a new video generative model performance metric that captures semantics more directly than the visual quality and comparison metrics currently in widespread use.

## 2   Sampling long videos

Our goal in this paper is to sample coherent photo-realistic videos $\mathbf{v}$ with thousands of frames (see Fig. 1). To sample an arbitrarily long video with a generative model that can sample or condition on only a small number of frames at once, we must use a sequential procedure. The simplest example of this is an autoregressive scheme, an example of which is shown in Fig. 2a for a video completion task. In this example it takes seven stages to sample a complete video, in that we must run the generative model's sampling procedure seven times. At each stage three frames are sampled conditioned on the immediately preceding four frames. This scheme is appealing for its simplicity but imposes a strong assumption that, given the set of four frames that are conditioned on at a particular stage, all frames that come afterwards are conditionally independent of all frames that came before. This restriction can be partially ameliorated with the sampling scheme shown in Fig. 2b where, in the first three stages, every second frame is sampled and then, in the remaining four stages, the remaining frames are infilled. One way to implement this would be to train two different models operating at the two different temporal resolutions. In the language of Ho et al. [16], who use a similar approach, sampling would be carried out in the first three stages by a "frameskip-2" model and, in the remaining stages, by a "frameskip-1" model. Both this approach and the autoregressive approach are examples of what we call *sampling schemes*. More generally, we characterize a sampling scheme as a sequence of tuples $[(\mathcal{X}_s, \mathcal{Y}_s)]_{s=1}^S$, each containing a vector $\mathcal{X}_s$ of indices of frames to sample and a vector $\mathcal{Y}_s$ of indices of frames to condition on for stages $s = 1, \ldots, S$.

**Algorithm 1** Sample a video $\mathbf{v}$ given a sampling scheme $[(\mathcal{X}_s, \mathcal{Y}_s)]_{s=1}^{S}$. For unconditional generation, the input $\mathbf{v}$ can be a tensor of zeros. For conditional generation, the observed input frames should contain their observed values.

1: **procedure** SAMPLEVIDEO($\mathbf{v}; \theta$)
2:     **for** $s \leftarrow 1, \ldots, S$ **do**
3:         $\mathbf{y} \leftarrow \mathbf{v}[\mathcal{Y}_s]$                                              ▷ Gather frames indexed by $\mathcal{Y}_s$.
4:         $\mathbf{x} \sim \text{DDPM}(\cdot; \mathbf{y}, \mathcal{X}_s, \mathcal{Y}_s, \theta)$                         ▷ Sample $\mathbf{x}$ from the conditional DDPM.
5:         $\mathbf{v}[\mathcal{X}_s] \leftarrow \mathbf{x}$                 ▷ Modify frames indexed by $\mathcal{X}_s$ with their sampled values.
6: **return** $\mathbf{v}$

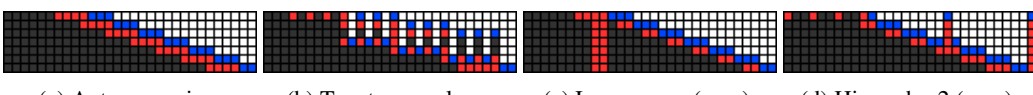

| (a) Autoregressive. | (b) Two temporal res. | (c) Long-range (ours). | (d) Hierarchy-2 (ours). |

Figure 2: Sampling schemes to complete a video of length $N = 30$ conditioned on the first 10 frames, with access to at most $K = 7$ frames at a time. Each stage $s$ of the sampling procedure is represented by one row in the figure, going from top to bottom. Within each subfigure, one column represents one frame of the video, from frame one on the left to frame 30 on the right. At each stage, the values of frames marked in blue are sampled conditioned on the (observed or previously sampled) values of frames marked in red; frames marked in gray are ignored; and frames marked in white are yet to be sampled. For every sampling scheme, all video frames have been sampled after the final row.

Algorithm 1 lays out how such a sampling scheme is used to sample a video. If the underlying generative model is trained specifically to model sequences of consecutive frames, or sequences of regularly-spaced frames, then the design space for sampling schemes compatible with these models is severely constrained. In this paper we take a different approach. We design and train a generative model to sample any arbitrarily-chosen subset of video frames conditioned on any other subset and train it using an entirely novel distribution of such tasks. In short, our model is trained to generate frames for any choice of $\mathcal{X}$ and $\mathcal{Y}$. The only constraint we impose on our sampling schemes is therefore a computational consideration that $|\mathcal{X}_s| + |\mathcal{Y}_s| \leq K$ for all $s$ but, to generate meaningful videos, any valid sampling scheme must also satisfy two more constraints: (1) all frames are sampled at at least one stage and (2) frames are never conditioned upon before they are sampled.

Such a flexible generative model allows us to explore and use sampling schemes like those in Fig. 2c and Fig. 2d. We find in our experiments that the best video sampling scheme is dataset dependent. Accordingly, we have developed methodology to optimize such sampling schemes in a dataset dependent way, leading to improved video quality as measured by the Fréchet Video Distance [33] among other metrics. We now review conditional DDPMs (Section 3), before discussing the FDM's architecture, the specific task distribution used to train it, and the choice and optimization of sampling schemes in Section 4.

## 3 A review of conditional denoising diffusion probabilistic models

Denoising diffusion probabilistic models, or DDPMs [28, 15, 22, 30], are a class of generative model for data $\mathbf{x}$, which throughout this paper will take the form of a 4-dimensional tensor representing multiple video frames. We will describe the conditional extension [32], in which the modeled $\mathbf{x}$ is conditioned on observations $\mathbf{y}$. DDPMs simulate a diffusion process which transforms $\mathbf{x}$ to noise, and generate data by learning the probabilistic inverse of the diffusion process. The diffusion process happens over timesteps $0, \ldots, T$ such that $\mathbf{x}_0 = \mathbf{x}$ is data without noise, $\mathbf{x}_1$ has a very small amount of noise added, and so on until $\mathbf{x}_T$ is almost independent of $\mathbf{x}_0$ and approximates a random sample from a unit Gaussian. In the diffusion process we consider, the distribution over $\mathbf{x}_t$ depends only on $\mathbf{x}_{t-1}$:

$$q(\mathbf{x}_t|\mathbf{x}_{t-1}) = \mathcal{N}(\mathbf{x}_t; \sqrt{\alpha_t}\mathbf{x}_{t-1}, (1 - \alpha_t)\mathbf{I}). \tag{1}$$

Hyperparameters $\alpha_1, \ldots, \alpha_T$ are chosen to all be close to but slightly less than 1 so that the amount of noise added at each step is small. The combination of this diffusion process and a data distribution

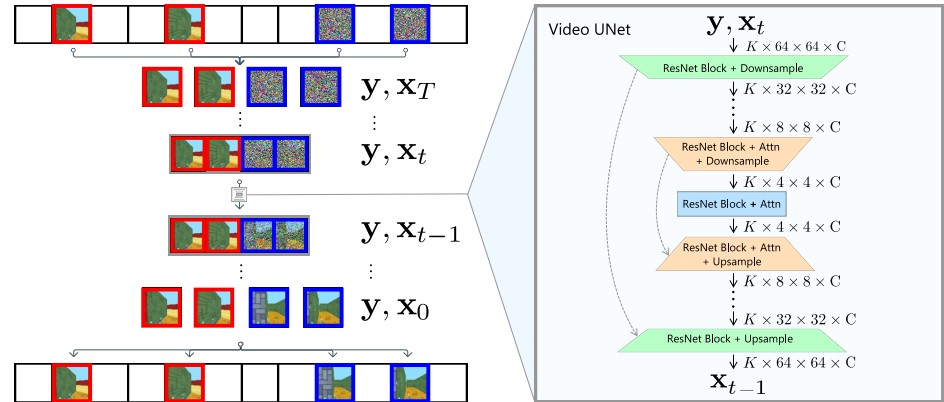

**Figure 3: Left:** Our DDPM iteratively transforms Gaussian noise $\mathbf{x}_T$ to video frames $\mathbf{x}_0$ (shown with blue borders), conditioning on observed frames $\mathbf{y}$ (red borders) at every step. **Right:** The U-net architecture used within each DDPM step. It computes $\epsilon_\theta(\mathbf{x}_t, \mathbf{y}, t)$, with which the Gaussian transition $p_\theta(\mathbf{x}_{t-1}|\mathbf{x}_t)$ is parameterized.

$q(\mathbf{x}_0, \mathbf{y})$ (recalling that $\mathbf{x}_0 = \mathbf{x}$) defines the joint distribution

$$q(\mathbf{x}_{0:T}, \mathbf{y}) = q(\mathbf{x}_0, \mathbf{y}) \prod_{t=1}^{T} q(\mathbf{x}_t|\mathbf{x}_{t-1}). \tag{2}$$

DDPMs work by "inverting" the diffusion process: given values of $\mathbf{x}_t$ and $\mathbf{y}$ a neural network is used to parameterize $p_\theta(\mathbf{x}_{t-1}|\mathbf{x}_t, \mathbf{y})$, an approximation of $q(\mathbf{x}_{t-1}|\mathbf{x}_t, \mathbf{y})$. This neural network lets us draw samples of $\mathbf{x}_0$ by first sampling $\mathbf{x}_T$ from a unit Gaussian (recall that the diffusion process was chosen so that $q(\mathbf{x}_T)$ is well approximated by a unit Gaussian), and then iteratively sampling $\mathbf{x}_{t-1} \sim p_\theta(\cdot|\mathbf{x}_t, \mathbf{y})$ for $t = T, T-1, \ldots, 1$. The joint distribution of sampled $\mathbf{x}_{0:T}$ given $\mathbf{y}$ is

$$p_\theta(\mathbf{x}_{0:T}|\mathbf{y}) = p(\mathbf{x}_T) \prod_{t=1}^{T} p_\theta(\mathbf{x}_{t-1}|\mathbf{x}_t, \mathbf{y}) \tag{3}$$

where $p(\mathbf{x}_T)$ is a unit Gaussian that does not depend on $\theta$. Training the conditional DDPM therefore involves fitting $p_\theta(\mathbf{x}_{t-1}|\mathbf{x}_t, \mathbf{y})$ to approximate $q(\mathbf{x}_{t-1}|\mathbf{x}_t, \mathbf{y})$ for all choices of $t$, $\mathbf{x}_t$, and $\mathbf{y}$.

Several observations have been made in recent years which simplify the learning of $p_\theta(\mathbf{x}_{t-1}|\mathbf{x}_t, \mathbf{y})$. Sohl-Dickstein et al. [28] showed that when $\alpha_t$ is close to 1, $p_\theta(\mathbf{x}_{t-1}|\mathbf{x}_t)$ is approximately Gaussian [28]. Furthermore, Ho et al. [15] showed that this Gaussian's variance can be modeled well with a non-learned function of $t$, and that a good estimate of the Gaussian's mean can be obtained from a "denoising model" as follows. Given data $\mathbf{x}_0$ and unit Gaussian noise $\epsilon$, the denoising model (in the form of a neural network) is fed "noisy" data $\mathbf{x}_t := \sqrt{\tilde{\alpha}_t}\mathbf{x}_0 + \sqrt{1 - \tilde{\alpha}_t}\epsilon$ and trained to recover $\epsilon$ via a mean squared error loss. The parameters $\tilde{\alpha}_t := \prod_{i=1}^{t} \alpha_i$ are chosen to ensure that the marginal distribution of $\mathbf{x}_t$ given $\mathbf{x}_0$ is $q(\mathbf{x}_t|\mathbf{x}_0)$ as derived from Eq. (1). Given a weighting function $\lambda(t)$, the denoising loss is

$$\mathcal{L}(\theta) = \mathbb{E}_{q(\mathbf{x}_0, \mathbf{y}, \epsilon)} \left[ \sum_{t=1}^{T} \lambda(t)\|\epsilon - \epsilon_\theta(\mathbf{x}_t, \mathbf{y}, t)\|_2^2 \right] \quad \text{with} \quad \mathbf{x}_t = \sqrt{\tilde{\alpha}_t}\mathbf{x}_0 + \sqrt{1 - \tilde{\alpha}_t}\epsilon. \tag{4}$$

The mean of $p_\theta(\mathbf{x}_{t-1}|\mathbf{x}_t, \mathbf{y})$ is obtained from the denoising model's output $\epsilon_\theta(\mathbf{x}_t, \mathbf{y}, t)$ as $\frac{1}{\alpha_t}\mathbf{x}_t - \frac{1-\alpha_t}{\sqrt{1-\tilde{\alpha}_t}}\epsilon_\theta(\mathbf{x}_t, \mathbf{y}, t)$. If the weighting function $\lambda(t)$ is chosen appropriately, optimising Eq. (4) is equivalent to optimising a lower-bound on the data likelihood under $p_\theta$. In practice, simply setting $\lambda(t) := 1$ for all $t$ can produce more visually compelling results in the image domain [15].

In our proposed method, as in Tashiro et al. [32], the shapes of $\mathbf{x}_0$ and $\mathbf{y}$ sampled from $q(\cdot)$ vary. This is because we want to train a model which can flexibly adapt to e.g. varying numbers of observed frames. To map Eq. (4) to this scenario, note that both $\mathbf{x}_0$ and $\mathbf{y}$ implicitly contain information about which frames in the video they represent (via the index vectors $\mathcal{X}$ and $\mathcal{Y}$ introduced in the previous section). This information is used inside the neural network $\epsilon_\theta(\mathbf{x}_t, \mathbf{y}, t)$ so that interactions between frames can be conditioned on the distance between them (as described in the following section) and also to ensure that the sampled noise vector $\epsilon$ has the same shape as $\mathbf{x}_0$.

**Algorithm 2** Sampling training tasks $\mathcal{X}, \mathcal{Y} \sim u(\cdot)$ given $N, K$.

1:  $\mathcal{X} := \{\}; \mathcal{Y} := \{\}$
2:  **while** True **do**
3:     $n_{\text{group}} \sim \text{UniformDiscrete}(1, K)$
4:     $s_{\text{group}} \sim \text{LogUniform}(1, (N-1)/n_{\text{group}})$
5:     $x_{\text{group}} \sim \text{Uniform}(0, N - (n_{\text{group}} - 1) \cdot s_{\text{group}})$
6:     $o_{\text{group}} \sim \text{Bernoulli}(0.5)$
7:     $\mathcal{G} := \{\lfloor x_{\text{group}} + s_{\text{group}} \cdot i \rfloor | i \in \{0, \ldots, n_{\text{group}} - 1\}\} \setminus \mathcal{X} \setminus \mathcal{Y}$
8:     **if** $|\mathcal{X}| + |\mathcal{Y}| + |\mathcal{G}| > K$ **then**
9:         **return** $\texttt{set2vector}(\mathcal{X}), \texttt{set2vector}(\mathcal{Y})$
10:    **else if** $|\mathcal{X}| = 0$ **or** $o_{\text{group}} = 0$ **then**
11:       $\mathcal{X} := \mathcal{X} \cup \mathcal{G}$
12:    **else**
13:       $\mathcal{Y} := \mathcal{Y} \cup \mathcal{G}$

Figure 4: **Left:** Samples from $u(\mathcal{X}, \mathcal{Y})$ with video length $N = 30$ and limit $K = 10$ on the number of sampled indices. Each row shows one sample and columns map to frames, with frame 1 on the left and frame $N$ on the right. Blue and red denote latent and observed frames respectively. All other frames are ignored and shown as white. **Right:** Pseudocode for drawing these samples. The while loop iterates over a series of regularly-spaced groups of latent variables. Each group is parameterized by: the number of indices in it, $n_{\text{group}}$; the spacing between indices in it, $s_{\text{group}}$; the position of the first frame in it, $x_{\text{group}}$, and an indicator variable for whether this group is observed, $o_{\text{group}}$ (which is ignored on line 10 if $\mathcal{X}$ is empty to ensure that the returned value of $\mathcal{X}$ is never empty). These quantities are sampled in a continuous space and then discretized to make a set of integer coordinates on line 7. The process repeats until a group is sampled which, if added to $\mathcal{X}$ or $\mathcal{Y}$, will cause the number of frames to exceed $K$. That group is then discarded and $\mathcal{X}$ and $\mathcal{Y}$ are returned as vectors. The FDM's training objective forces it to work well for any $(\mathcal{X}, \mathcal{Y})$ pair from this broad distribution.

## 4 Training procedure and architecture

**Training task distribution** Different choices of latent and observed indices $\mathcal{X}$ and $\mathcal{Y}$ can be regarded as defining different conditional generation tasks. In this sense, we aim to learn a model which can work well on any task (i.e. any choice of $\mathcal{X}$ and $\mathcal{Y}$) and so we randomly sample these vectors of indices during training. We do so with the distribution $u(\mathcal{X}, \mathcal{Y})$ described in Fig. 4. This provides a broad distribution covering many plausible test-time use cases while still providing sufficient structure to improve learning (see ablation in Section 6 and more details in Appendix C). To cope with constrained computational resources, the distribution is designed such that $|\mathcal{X}| + |\mathcal{Y}|$ is upper-bounded by some pre-specified $K$. Sampling from $q(\mathbf{x}_0, \mathbf{y})$ in Eq. (4) is then accomplished by randomly selecting both a full training video $\mathbf{v}$ and indices $\mathcal{X}, \mathcal{Y} \sim u(\cdot, \cdot)$. We then extract the specified frames $\mathbf{x} = \mathbf{v}[\mathcal{X}]$ and $\mathbf{y} = \mathbf{v}[\mathcal{Y}]$ (where we use $\mathbf{v}[\mathcal{X}]$ to denote the concatenation of all frames in $\mathbf{v}$ with indices in $\mathcal{X}$ and and $\mathbf{v}[\mathcal{Y}]$ similarly).

**Architecture** DDPM image models [15, 22] typically use a U-net architecture [24]. Its distinguishing feature is a series of spatial downsampling layers followed by a series of upsampling layers, and these are interspersed with convolutional res-net blocks [14] and spatial attention layers. Since we require an architecture which operates on 4-D video tensors rather than 3-D image tensors we add an extra *frame* dimension to its input, output and hidden state, resulting in the architecture shown on the right of Fig. 3. We create the input to this architecture as a concatenation $\mathbf{x}_t \oplus \mathbf{y}$, adding an extra input channel which is all ones for observed frames and all zeros for latent frames. For RGB video, the input shape is therefore $(K, \textit{image height}, \textit{image width}, 4)$. Since the output should have the same shape as $\mathbf{x}_t$ we only return outputs corresponding to the latent frames, giving output shape $(|\mathcal{X}|, \textit{image height}, \textit{image width}, 3)$. We run all layers from the original model (including convolution, resizing, group normalization, and spatial attention) independently for each of the $K$ frames. To allow communication between the frames, we add a temporal attention layer after each spatial attention layer, described in more detail in the appendix. The spatial attention layer allows each spatial location to attend to all other spatial locations *within* the same frame, while the temporal attention layer allows each spatial location to attend to the same spatial location across all *other* frames. This combination of a temporal attention layer with a spatial attention layer is sometimes

referred to as *factorized attention* [32, 16]. We found that, when using this architecture in conjunction with our meta-learning approach, performance could be improved by using a novel form of relative position encoding [27, 38]. This is included in our released source code but we leave its exposition to the supplementary material.

**Training batch padding**   Although the size $|\mathcal{X} \oplus \mathcal{Y}|$ of index vectors sampled from our training distribution is bounded above by $K$, it can vary. To fit examples with various sizes of index vectors into the same batch, one option would be to pad them all to length $K$ with zeros and use masks so that the zeros cannot affect the loss. This, however, would waste computation on processing tensors of zeros. We instead use this computation to obtain a lower-variance loss estimate by processing additional data with "training batch padding". This means that, for training examples where $|\mathcal{X} \oplus \mathcal{Y}| < K$, we concatenate frames uniformly sampled from a second video to increase the length along the frame-dimension to $K$. Masks are applied to the temporal attention mechanisms so that frames from different videos cannot attend to eachother and the output for each is the same as that achieved by processing the videos in different batches.

**Sampling schemes**   Before describing the sampling schemes we explore experimentally, we emphasize that the relative performance of each is dataset-dependent and there is no single best choice. A central benefit of FDM is that it can be used at test-time with different sampling schemes without retraining. Our simplest sampling scheme, **Autoreg**, samples ten consecutives frames at each stage conditioned on the previous ten frames. **Long-range** is similar to Autoreg but conditions on only the five most recent frames as well as five of the original 36 observed frames. **Hierarchy-2** uses a multi-level sampling procedure. In the first level, ten evenly spaced frames spanning the non-observed portion of the video are sampled (conditioned on ten observed frames). In the second level, groups of consecutive frames are sampled conditioned on the closest past and future frames until all frames have been sampled. **Hierarchy-3** adds an intermediate stage where several groups of variables with an intermediate spacing between them are sampled. We include adaptive hierarchy-2, abbreviated **Ad. hierarchy-2**, as a demonstration of a sampling scheme only possible with a model like FDM. It samples the same frames at each stage as Hierarchy-2 but selects which frames to condition on adaptively at test-time with a heuristic aimed at collecting the maximally diverse set of frames, as measured by the pairwise LPIPS distance [41] between them.

**Optimizing sampling schemes**   An appealing alternative to the heuristic sampling schemes described in the previous paragraph would be to find a sampling scheme that is, in some sense, optimal for a given model and video generation/completion task. While it is unclear how to tractably choose which frames should be sampled at each stage, we suggest that the frames to condition on at each stage can be chosen by greedily optimizing the diffusion model loss which, as mentioned in Section 3, is closely related to the data log-likelihood. Given a fixed sequence of frames to sample at each stage $[\mathcal{X}_s]_{s=1}^S$ we select $\mathcal{Y}_s$ for each $s$ to minimize Eq. (4). This is estimated using a set of 100 training videos and by iterating over 10 evenly-spaced values of $t$ (which reduced variance relative to random sampling of $t$). See the appendix for further details. We create two optimized sampling schemes: one with the same latent indices as Autoreg, and one with the same latent indices as Hierarchy-2. We call the corresponding optimized schemes **Opt. autoreg** and **Opt. hierarchy-2**.

## 5   CARLA Town01 Dataset

In addition to our methodological contributions, we propose a new video-modeling dataset and benchmark which provides an interpretable measure of video completion quality. The dataset consists of videos of a car driving with a first-person view, produced using the CARLA autonomous driving simulator [9]. All 408 training and 100 test videos (of length 1000 frames and resolution $128 \times 128$) are produced within a single small town, CARLA's Town01. As such, when a sufficiently expressive video model is trained on this dataset it memorizes the layout of the town and videos sampled from the model will be recognisable as corresponding to routes travelled within the town. We train a regression model in the form of a neural network which maps with high accuracy from any single rendered frame to $(x, y)$ coordinates representing the car's position. Doing so allows us to plot the routes corresponding to sampled videos (see left of Fig. 5) and compute semantically-meaningful yet quantitative measures of the validity of these routes. Specifically, we compute histograms of speeds, where each speed is estimated by measuring the distance between the regressed locations for frames

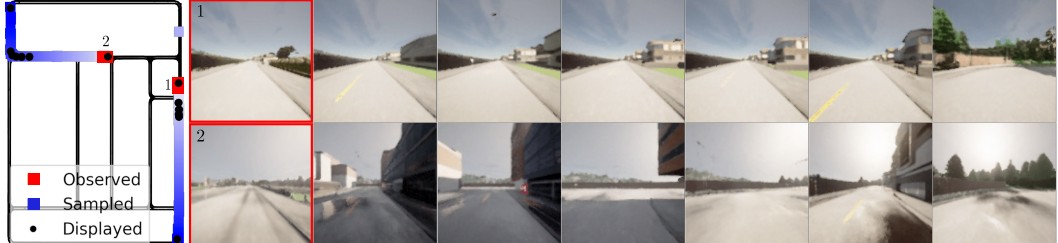

Figure 5: **Left:** Map of the town featured in the CARLA Town01 dataset. We visualize two video completions by FDM by showing coordinates output by our regressor (discussed in Section 5) for each frame. Those corresponding to the initial 36 observed frames are shown in red and those for the 964 sampled frames are shown in blue. **Right:** For each completion, we show one of the initially observed frames followed by four of the sampled frames (at positions chosen to show the progression with respect to visible landmarks and marked by black dots on the map). The town's landmarks are usually sampled with high-fidelity, which is key to allowing the regressor to produce a coherent trajectory on the left. However there are sometimes failures: a blue square near the top-right of the map shows where the video model "jumped" to a wrong location for a single frame.

spaced ten apart (1 second at the dataset's frame rate). Sampled videos occasionally "jump" between disparate locations in the town, resulting in unrealistically large estimated speeds. To measure the frequency of these events for each method, we compute the percentage of our point-speed estimates that exceed a threshold of 10m/s (the dataset was generated with a maximum simulated speed of 3m/s). We report this metric as the outlier percentage (OP). After filtering out these outliers, we compute the Wasserstein distance (WD) between the resulting empirical distribution and that of the original dataset, giving a measure of how well generated videos match the speed of videos in the dataset. We release the CARLA Town 01 dataset along with code and our trained regression model to allow future comparisons.[2]

## 6    Experiments

We perform our main comparisons on the video completion task. In keeping with Saxena et al. [26], we condition on the first 36 frames of each video and sample the remainder. We present results on three datasets: GQN-Mazes [10], in which videos are 300 frames long; MineRL Navigate [13, 26] (which we will from now on refer to as simply MineRL), in which videos are 500 frames long; and the CARLA Town01 dataset we release, for which videos are 1000 frames long. We train FDM in all cases with the maximum number of represented frames $K = 20$. We host non-cherry-picked video samples (both conditional and unconditional) from FDM and all baselines online[3].

**Comparison of sampling schemes**    The relative performance of different sampling schemes varies significantly between datasets as shown in Table 1. We report Fréchet Video Distances (FVDs) [33], a measure of how similar sampled completions are to the test set, on all datasets. In addition on GQN-Mazes we report the accuracy metric [26], which classifies videos based on which rooms are visited and measures how often a completion is given the same class as the corresponding test video. For CARLA Town01 we report the previously described percentage outliers (PO) and Wasserstein distance (WD) metrics.

We can broadly consider the aforementioned sampling schemes as either being in the "autoregressive" family (Autoreg and Long-range) or in the "hierarchical"

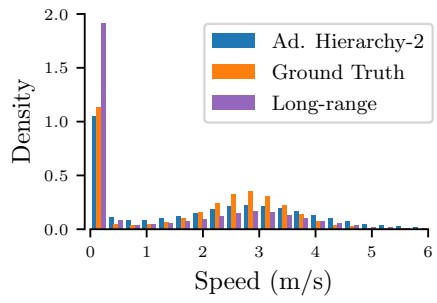

Figure 6: Speed distributions measured from sampled and ground-truth dataset videos.

---

[2]https://github.com/plai-group/flexible-video-diffusion-modeling
[3]https://www.cs.ubc.ca/~wsgh/fdm

Table 1: Evaluation on video completion with various modes of our method along with several baselines from the literature. Error bars denote the standard error computed with 5 random seeds. Higher is better for the accuracy metric [26] and lower is better for all other metrics shown.

| Model | Sampling scheme | GQN-Mazes | | MineRL | CARLA Town01 | | |
| | | FVD | Accuracy | FVD | FVD | WD | OP |
|---|---|---|---|---|---|---|---|
| CWVAE [26] | CWVAE | $837 \pm 8$ | $82.6 \pm 0.5$ | $1573 \pm 5$ | 1161 | 0.666 | 44.4 |
| TATS [11] | TATS | $163 \pm 2.6$ | $77.0 \pm 0.8$ | $807 \pm 14$ | 329 | 1.648 | 42.4 |
| VDM [16] | VDM | $66.7 \pm 1.5$ | $77.8 \pm 0.5$ | $271 \pm 8.8$ | 169 | 0.501 | 16.9 |
| FDM (ours) | Autoreg | $86.4 \pm 5.2$ | $69.6 \pm 1.3$ | $281 \pm 10$ | 222 | 0.579 | 0.51 |
| | Long-range | $64.5 \pm 1.9$ | $77.0 \pm 1.4$ | $\mathbf{267 \pm 4.0}$ | 213 | 0.653 | **0.47** |
| | Hierarchy-2 | $\mathbf{53.1 \pm 1.1}$ | $82.8 \pm 0.7$ | $275 \pm 7.7$ | 120 | 0.318 | 3.28 |
| | Hierarchy-3 | $53.7 \pm 1.9$ | $\mathbf{83.8 \pm 1.1}$ | $311 \pm 6.8$ | 149 | 0.363 | 4.53 |
| | Ad. hierarchy-2 | $55.0 \pm 1.4$ | $83.2 \pm 1.3$ | $316 \pm 8.9$ | **117** | **0.311** | 3.44 |

family (the remainder). Those in the hierarchical family achieve significantly better FVDs [33] on GQN-Mazes. Our samples in the appendix suggest that this is related to the autoregressive methods "forgetting" the colors of walls after looking away from them for a short time. In contrast, for MineRL the autoregressive methods tend to achieve the best FVDs. This may relate to the fact that trajectories in MineRL tend to travel in straight lines through procedurally-generated "worlds"[13, 26], limiting the number of long-range dependencies. Finally on CARLA Town01 we notice qualitatively different behaviours from our autoregressive and hierarchical sampling schemes. The hierarchical sampling schemes have a tendency to occasionally lose coherence and "jump" to different locations in the town. This is reflected by higher outlier percentages (OP) in Table 1. On the other hand the autoregressive schemes often stay stationary for unrealistically long times at traffic lights. This is reflected in the histogram of speeds in Fig. 6, which has a larger peak around zero than the ground truth. The high variance of the sampling scheme's relative performance over different datasets points to a strength of our method, which need only be trained once and then used to explore a variety of sampling schemes. Furthermore, we point out that the best FVDs in Table 1 on all datasets were obtained using sampling schemes that could not be implemented using models trained in prior work, or over evenly spaced frames.

**Comparison with baselines** The related work most relevant to ours is the concurrent work of Ho et al. [16], who model 64-frame videos using two trained DDPMs. The first is a "frameskip-4" model trained to generate every fourth frame and the second is a "frameskip-1" model trained on sequences of nine consecutive frames and used to "fill in" the gaps between frames generated in the first stage. To compare against this approach, which we denote **VDM**, we train both a "frameskip-4" and a "frameskip-1" model with architectures identical to our own.[4] Since VDM requires two trained DDPMs, we train it for more GPU-hours than FDM despite the fact that FDM is meta-learning over a far broader task distribution. We also compare against **TATS** [11], which embeds videos into a discrete latent space before modelling them with a transformers, and the clockwork VAE (**CWVAE**) [26], a VAE-based model specifically designed to maintain long-range dependencies within video.

Both the diffusion-based methods, FDM and VDM, achieve significantly higher FVD scores than TATS and CWVAE. This may point toward the utility of diffusion models in general for modeling images and video. Table 1 also makes clear the main benefit of FDM over VDM: although there is no sampling scheme for FDM which always outperforms VDM, there is at least one sampling scheme that outperforms it on each dataset. This speaks to the utility of learning a flexible model like FDM that allows different sampling schemes to be experimented with after training.

**Optimized sampling schemes** As mentioned in Section 4, another advantage of FDM is that it makes possible a model- and dataset-specific optimization procedure to determine on which frames

---

[4]The VDM is concurrent work and, at the time of writing, without a code-release. Since we intend this primarily as a comparison against the VDM sampling scheme we do not reimplement their exact architecture and note that there are other differences including their approach to imputation.

Table 2: FVD scores for our sampling schemes with observed indices optimized offline as described in Section 4. We mark with an asterisk (*) the eight numbers which improve on the corresponding non-optimized sampling schemes and highlight in bold those that are better than any in Table 1.

| | GQN-Mazes | | MineRL | CARLA Town01 | | |
|---|---|---|---|---|---|---|
| Sampling scheme | FVD | Accuracy | FVD | FVD | WD | OP |
| Opt. autoreg | $53.6 \pm 1.2^*$ | $80.2 \pm 1.2^*$ | $\mathbf{257 \pm 6.8}^*$ | $146^*$ | $0.452^*$ | $0.65$ |
| Opt. hierarchy-2 | $\mathbf{51.1 \pm 1.3}^*$ | $\mathbf{84.6 \pm 0.7}^*$ | $320 \pm 7.0$ | $124$ | $0.349$ | $4.11^*$ |

to condition. Table 2 shows the results when this procedure is used to create sampling schemes for different datasets. In the first row we show results where the latent frames are fixed to be those of the Autoreg sampling scheme, and in the second row the latent frames are fixed to match those of Hierarchy-2. On two of the three datasets the best results in Table 1 are improved upon, showing the utility of this optimization procedure.

**Comparison with training on a single task**   Training a network with our distribution over training tasks could be expected to lead to worse performance on a single task than training specifically for that task. To test whether this is the case, we train an ablation of FDM with training tasks exclusively of the type used in our Autoreg sampling scheme, i.e. "predict ten consecutive frames given the previous ten." Tested with the Autoreg sampling scheme, it obtained an FVD of 82.0 on GQN-Mazes and 234 on MineRL. As expected given the specialization to a single task, this is better than when FDM is run with the Autoreg sampling scheme (obtaining FVDs of 86.4 and 281 respectively).

**Ablation on training task distribution**   To test how important our proposed structured training distribution is to FDM's performance, we perform an ablation with a different task distribution that samples $\mathcal{X}$ and $\mathcal{Y}$ from uniform distributions instead of our proposed structured task distribution We provide full details in the appendix, but report here that switching away form our structured training distribution made the FVD scores worse on all five tested sampling schemes on both GQN-Mazes and MineRL. The reduction in the average FVD was $31\%$ on GQN-Mazes and $52\%$ on MineRL. This implies that our structured training distribution has a significant positive effect.

# 7   Related work

Some related work creates conditional models by adapting the sampling procedure of an *unconditional* DDPM [30, 18, 21, 16]. These approaches require approximations and the more direct approach that we use (explcitly training a conditional DDPM) was shown to have benefits by Tashiro et al. [32]. We consider further comparison of these competing approaches to be outside the focus of this work, which is on modeling a small portion of video frames at a time, essentially performing marginalization in addition to conditioning.

There are a number of approaches in the literature which use VAEs rather than DDPMs for video modelling. Babaeizadeh et al. [2] use a VAE model which predicts frames autoregressively conditioned on a global time-invariant latent variable. A related approach by Denton and Fergus [7] also uses a VAE with convolutional LSTM architectures in both the encoder and decoder. Unlike Babaeizadeh et al. [2] the prior is learned and a different latent variable is sampled for each frame. Babaeizadeh et al. [3] use a VAE with one set of latent variables per frame and inter-frame dependencies tracked by a two-layer LSTM. Their architecture intentionally overfits to the training data, which when coupled with image augmentations techniques achieves SOTA on various video prediction tasks. Kim et al. [20] use a variational RNN [5] with a hierarchical latent space that includes binary indicator variables which specify how the video is divided into a series of subsequences. Both Villegas et al. [35] and Wichers et al. [37] target long-term video prediction using a hierarchical variational LSTM architecture, wherein high-level features such as landmarks are predicted first, then decoded into low-level pixel space. The two approaches differ in that Villegas et al. [35] requires ground truth landmark labels, while [37] removes this dependence using an unsupervised adversarial approach. Fully GAN-based video models have also been proposed [1, 6] but generally suffer from "low quality frames or low number of frames or both" [1].

# 8 Discussion

We have defined and empirically explored a new method for generating photorealistic videos with long-range coherence that respects and efficiently uses fixed, finite computational resources. Our approach outperforms prior work on long-duration video modeling as measured by quantitative and semantically meaningful metrics and opens up several avenues for future research. For one, similar to using DDPMs for image generation, our method is slow to sample from (it takes approximately 16 minutes to generate a 300 frame video on a GPU). Ideas for making sampling faster by decreasing the number of integration steps [25, 29, 39] could be applied to our video model.

On a different note, consider the datasets on which our artifact was trained. In each there was a policy for generating the sequences of actions that causally led to the frame-to-frame changes in camera pose. In MineRL the video was generated by agents that were trained to explore novel Minecraft worlds to find a goal block approximately 64 meters away [13]. The CARLA data was produced by a camera attached to an agent driven by a low level proportional–integral–derivative controller following waypoints laid down by a high level planner that was given new, random location goals to drive to intermittently. In both cases our video model had no access to either the policy or the specific actions taken by these agents and, so, in a formal sense, our models integrate or marginalize over actions drawn from the stochastic policy used to generate the videos in the first place. Near-term future work could involve adding other modalities (e.g. audio) to FDM as well as explicitly adding actions and rewards, transforming our video generative model into a vision-based world model in the reinforcement learning sense [17, 19]. Furthermore, we point out that FDM trained on CARLA Town01 is in theory capable of creating 100-second videos conditioned on both the first and final frame. Doing so can be interpreted as running a "visual" controller which proposes a path between a current state and a specified goal. Preliminary attempts to run in FDM in this way yielded inconsistent results but we believe that this could be a fruitful direction for further investigation.

**Acknowledgments**

We would like to thank Inverted AI, and especially Alireza Morsali, for generating the CARLA Town01 dataset. We acknowledge the support of the Natural Sciences and Engineering Research Council of Canada (NSERC), the Canada CIFAR AI Chairs Program, and the Intel Parallel Computing Centers program. Additional support was provided by UBC's Composites Research Network (CRN), and Data Science Institute (DSI). This research was enabled in part by technical support and computational resources provided by WestGrid (www.westgrid.ca), Compute Canada (www.computecanada.ca), and Advanced Research Computing at the University of British Columbia (arc.ubc.ca). WH acknowledges support by the University of British Columbia's Four Year Doctoral Fellowship (4YF) program.

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
