# A Experimental details

Table 3: Experimental details for all results reported. The GPUs referenced are all either NVIDIA RTX A5000s or NVIDIA A100s. In rows where GPU-hours are given as a range, different runs with identical settings took varying times due to varying performance of our computational infrastructure.

| Experiment | Method | Res. | Params (millions) | Batch size | GPUs | Iterations (thousands) | GPU-hours | K | Diffusion steps |
|---|---|---|---|---|---|---|---|---|---|
| GQN-Mazes | FDM | 64 | 78 | 8 | 1x A100 | 950 | 156 | 20 | 1000 |
| | VDM (frameskip-1/4) | 64 | 78/78 | 8/8 | 1x A100 | 1600/1100 | 151/153 (total 304) | N/A | 1000 |
| | TATS (VQGAN/Tran.) | 64 | 61/423 | 96/16 | 8/8x A100 | 72/1320 | 1314/1344 (total 2658) | N/A | N/A |
| | CWVAE | 64 | 34 | 50 | 1x A100 | 110 | 148 | N/A | N/A |
| MineRL | FDM | 64 | 78 | 8 | 1x A100 | 850 | 156 | 20 | 1000 |
| | VDM (frameskip-1/4) | 64 | 78/78 | 8/8 | 1x A100 | 1600/1100 | 161/163 (total 324) | N/A | 1000 |
| | TATS (VQGAN/Tran.) | 64 | 61/423 | 96/16 | 8/8x A100 | 72/550 | 1328/1056 (total 2384) | N/A | N/A |
| | CWVAE | 64 | 34 | 50 | 1x A100 | 60 | 41 | N/A | N/A |
| CARLA Town01 | FDM | 128 | 80 | 8 | 4x A100 | 500 | 380 | 20 | 1000 |
| | VDM (frameskip-1/4) | 128 | 80/80 | 4/4 | 2/4x A100 | 1750/1000 | 332/380 (total 712) | N/A | 1000 |
| | TATS (VQGAN/Tran.) | 128 | 61/423 | 48/8 | 4/4x A100 | 156/270 | 652/242 (total 894) | N/A | N/A |
| | CWVAE | 64 | 34 | 50 | 1x A100 | 70 | 115 | N/A | N/A |
| Ablations on GQN-Mazes | FDM (incl. ablations) | 64 | 78 | 3 | 1x A5000 | 500 | 40-70 | 10 | 250 |
| Ablations on MineRL | FDM (incl. ablations) | 64 | 78 | 3 | 1x A5000 | 500 | 40-70 | 10 | 250 |

The total compute required for this project, including all training, evaluation, and preliminary runs, was roughly 3.5 GPU-years. We used a mixture of NVIDIA RTX A5000s (on an internal cluster) and NVIDIA A100s (from a cloud provider).

Due to the expensive nature of drawing samples from both FDM and our baselines, we compute all quantitative metrics reported over the first 100 videos of the test set for GQN-Mazes and MineRL. For CARLA Town01, the test set length is 100. Table 3 lists the hyperparameters for all training runs reported. We provide additional details on the implementations of each method below.

**FDM** Our implementation of FDM builds on the DDPM implementation[5] of Nichol and Dhariwal [22]. For experiments at $64 \times 64$ resolution, the hyperparameters of our architecture are almost identical to that of their $64 \times 64$ image generation experiments: for example we use 128 as the base number of channels, the same channel multipliers at each resolution, and 4-headed attention. The exception is that we decrease the number of res-net blocks from 2 to 1 at each up/down-sampling step. As mentioned in the main text, we run all layers from the image DDPM independently and in parallel for each frame, and add a temporal attention layer after every spatial attention layer. The temporal attention layer has the same hyperparameters as the spatial attention layer (e.g. 4 attention heads) except for the addition of relative position encodings, which we describe later. For experiments at $128 \times 128$ resolution, we use almost the same architecture, but with an extra block at $128 \times 128$ resolution with channel multiplier 1. For full transparency, we release FDM's source code.

**VDM** As mentioned in the main paper, we train VDM by simply training two networks, each with architecture identical to that of FDM but different training tasks. In each of VDM's training tasks, we use a slice of 16 or 9 frames (with frameskip 4 or 1 respectively). We randomly sample zero or more "groups" of regularly-spaced frames to observe (where groups of frames are sampled similarly here to in FDM's structured mask distribution in Algorithm 2), and the rest are latent. On all datasets, we train each of the two networks forming the VDM baseline with roughly as many GPU-hours as FDM, so that VDM receives roughly twice as much training compute in total.

**TATS** We train TATS using the official implementation[6] along with its suggested hyperparameters. For GQN-Mazes and MineRL we train each stage for close to a week and, following Ge et al. [11], train them on 8 GPUs in parallel. For all datasets, the total training computation is multiple times that of FDM. In the included video samples from our TATS baseline, some artifacts are clearly visible. It may be that these could be removed with further hyperparameter tuning, but we did not pursue this.

---

[5] https://github.com/openai/improved-diffusion
[6] https://github.com/SongweiGe/TATS

Table 4: Additional metrics for evaluation on video completion. Lower is better for the test "Loss" and LPIPS. Higher is better for SSIM and PSNR.

| Model | Sampling scheme | GQN-Mazes | | | | MineRL | | | | CARLA Town01 | | |
|---|---|---|---|---|---|---|---|---|---|---|---|---|
| | | Loss | LPIPS | SSIM | PSNR | Loss | LPIPS | SSIM | PSNR | LPIPS | SSIM | PSNR |
| CWVAE [26] | CWVAE | – | 0.41 | **0.64** | 16.3 | – | 0.50 | **0.59** | **19.3** | 0.53 | 0.71 | 15.5 |
| TATS [11] | TATS | – | 0.40 | 0.59 | 15.5 | – | 0.42 | 0.45 | 17.0 | 0.40 | 0.68 | 13.9 |
| VDM [16] | VDM | **6.04** | 0.39 | 0.61 | 16.1 | **8.48** | 0.33 | 0.54 | 19.2 | 0.35 | 0.71 | 15.4 |
| FDM (ours) | Autoreg | 6.41 | 0.40 | 0.60 | 15.5 | 9.80 | **0.32** | 0.53 | 18.9 | 0.28 | 0.74 | 17.5 |
| | Long-range | 6.41 | **0.37** | 0.61 | 16.3 | 9.79 | **0.32** | 0.54 | 19.0 | **0.26** | **0.75** | **18.5** |
| | Hierarchy-2 | 6.40 | **0.37** | 0.61 | **16.4** | 9.75 | 0.33 | 0.54 | 19.0 | 0.29 | 0.73 | 17.2 |
| | Hierarchy-3 | 6.38 | 0.38 | 0.62 | **16.4** | 9.54 | 0.33 | 0.54 | 19.1 | 0.31 | 0.72 | 16.9 |
| | Ad. hierarchy-2 | 6.40 | **0.37** | 0.62 | **16.4** | 9.80 | 0.33 | 0.53 | 19.0 | 0.30 | 0.72 | 17.0 |

Notably, the datasets which we experiment on generally have a lower frame-rate than those used by Ge et al. [11], meaning that neighboring frames are more different and so potentially harder to model.

**CWVAE**    We train CWVAE using the official implementation[7] and use hyperparameters as close as possible to those used in the implementation by Saxena et al. [26]. We use 600 epochs to train CWVAE on MineRL, as suggested by Saxena et al. [26], and train it for more iterations on both other datasets. On CARLA Town01, since CWVAE is not implemented for $128 \times 128$ images, we downsample all train and test data to $64 \times 64$.

## A.1    Additional evaluation metrics

We report additional evaluation metrics in Table 4. The "Loss" refers to the average DDPM loss (Eq. (4)) over the test set, such that an appropriate choice of $\lambda(t)$ would yield the ELBO of the test videos under each model and sampling scheme although, as in our training loss, we use $\lambda(t) := 1$ to de-emphasise pixel-level detail. The commonly-used [26, 3] LPIPS, SSIM and PSNR metrics measure frame-wise distances between each generated frame around the ground-truth. To account for stochasticity in the task, $k$ video completions are generated for each test video and the smallest distance to the ground-truth is reported. We report them for completeness, but do not believe that SSIM and PSNR correlate well with video quality due to the stochastic nature of our datasets. For example, see the CWVAE videos on MineRL at `https://www.cs.ubc.ca/~wsgh/fdm` which obtain higher SSIM and PSNR than other methods despite being much blurrier. Since SSIM and PSNR are related to the mean-squared error in pixel space, they favor blurry samples over more realistic samples. While increasing $k$ should counteract this effect, the effectiveness of this scales poorly with video length and and so this made little difference in the datasets we consider.

## A.2    Ablation on training task distribution

We mention in the main paper that we perform an ablation on the training task distribution. FVD scores from this ablation are reported in Table 5. We sample from the baseline "uniform" task distribution as follows (where Uniform(a, b) should be understood to assign probability to all integers between $a$ and $b$ *inclusive*):

1. Sample $n_{total} \sim \text{Uniform}(1, K)$.
2. Assign $\mathcal{Z}$ to be a vector of $n_{total}$ integers sampled without replacement from $\{1, \ldots, K\}$.
3. Sample $n_{obs} \sim \text{Uniform}(0, n_{total} - 1)$.
4. Assign the first $n_{obs}$ entries in $\mathcal{Z}$ to $\mathcal{Y}$ and the remainder to $\mathcal{X}$.

This leads to a much less structured distribution than that described in Fig. 4. The network trained using our proposed task distribution obtains a better FVD than our ablation on all tested combinations of dataset and sampling scheme. Note that all networks in this experiment use the hyperparameters reported in the bottom two rows of Table 3, explaining the disparity between FVDs here and in Table 1.

---

[7]`https://github.com/vaibhavsaxena11/cwvae`

Table 5: Ablation for our training task distribution.

| | | FDM | Uniform |
|---|---|---|---|
| GQN-Mazes | Autoreg | **245** | 327 |
| | Hierarchy-2 | **235** | 279 |
| | Long-range | **198** | 281 |
| | Hierarchy-3 | **176** | 284 |
| | Ad. hierarchy-2 | **178** | 281 |
| | Average | **226** | 296 |
| MineRL | Autoreg | **465** | 672 |
| | Hierarchy-2 | **586** | 902 |
| | Long-range | **504** | 783 |
| | Hierarchy-3 | **515** | 970 |
| | Ad. hierarchy-2 | **613** | 990 |
| | Average | **518** | 786 |

# B   Relative position encodings

**Relative position encoding background and use-case**   Our temporal attention layer is run independently at every spatial location, allowing each spatial location in every frame to attend to its counterparts at the same spatial location in every other frame. That is, denoting the input to a temporal attention layer $\mathbf{z}^{\text{in}}$ and the output $\mathbf{z}^{\text{out}}$, we compute the $K \times C$ slice $\mathbf{z}^{\text{out}}_{:,h,w,:} = \text{attn}(\mathbf{z}^{\text{in}}_{:,h,w,:})$ for every spatial position $(h, w)$. To condition the temporal attention on the frame's positions within the video, we use relative position encodings (RPEs) [27, 38] for each pair of frames. Let $pos(i) = (\mathcal{X} \oplus \mathcal{Y})_i$ be a function mapping the index of a frame within $\mathbf{z}$ to its index within the full video $\mathbf{v}$. Then the encoding of the relative position of frames $i$ and $j$ depends only on $pos(i) - pos(j)$. We write this RPE as the set of three vectors $\mathbf{p}_{ij} = \{\mathbf{p}^Q_{ij}, \mathbf{p}^K_{ij}, \mathbf{p}^V_{ij}\}$ which are used in a modified form of dot-product attention (described in the following paragraph). Since $\mathbf{p}_{ij}$ must be created for every $(i, j)$ pair in a sequence, computing it adds a cost which scales as $\mathcal{O}(K^2)$ and prior work has attempted to minimize this cost by parametrizing $\mathbf{p}_{ij}$ with a simple learned look-up table (LUT) as $\mathbf{p}_{ij} := \text{LUT}(pos(i) - pos(j))$. In the next paragraph we describe our alternative to the LUT, but first we describe how the RPEs are used in either case. We use the RPEs in the same way as Shaw et al. [27]. As in a standard transformer [34], a sequence of input vectors $\mathbf{z}^{\text{in}}_1, \ldots, \mathbf{z}^{\text{in}}_K$ are transformed to queries, keys, and values via the linear projections $\mathbf{q}_i = W^Q \mathbf{z}_i$, $\mathbf{k}_i = W^K \mathbf{z}_i$, and $\mathbf{v}_i = W^V \mathbf{z}_i$ for $i = 1, \ldots, K$. Given the RPEs for all $(i, j)$ pairs, and marking the operations involving them in blue, the output of the attention block is

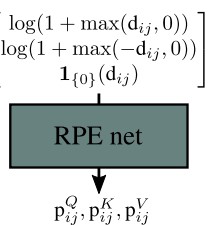

$$\begin{bmatrix} \log(1 + \max(\mathbf{d}_{ij}, 0)) \\ \log(1 + \max(-\mathbf{d}_{ij}, 0)) \\ \mathbf{1}_{\{0\}}(\mathbf{d}_{ij}) \end{bmatrix}$$

RPE net

$$\mathbf{p}^Q_{ij}, \mathbf{p}^K_{ij}, \mathbf{p}^V_{ij}$$

Figure 7: Parameterization of $f_{\text{RPE}}$ with $\mathbf{d}_{ij} := pos(i) - pos(j)$.

$$\mathbf{z}^{\text{out}}_i = \mathbf{z}^{\text{in}}_i + \sum_{j=1}^{K} \alpha_{ij}(\mathbf{v}_j + \mathbf{p}^V_{ij}) \quad \text{where} \quad \alpha_{ij} = \frac{\exp(e_{ij})}{\sum_{k=1}^{K} \exp(e_{ik})} \tag{5}$$

$$\text{with} \quad e_{ij} = \frac{1}{\sqrt{d_z}} \mathbf{q}_i^\mathsf{T} \mathbf{k}_j + \mathbf{p}^{Q\mathsf{T}}_{ij} \mathbf{k}_j + \mathbf{q}_i^\mathsf{T} \mathbf{p}^K_{ij}.$$

**Our approach to computing RPEs**   We argue that the simplicity of parametrizing RPEs with a LUT is not necessary within our framework for three reasons. **(1)** In our framework, $K$ can be kept small, so the $\mathcal{O}(K^2)$ scaling cost is of limited concern. **(2)** Furthermore, since the temporal attention mechanism is run for all spatial locations $(h, w)$ in parallel, the cost of computing RPEs can be shared between them. **(3)** The range of values that $pos(i) - pos(j)$ can take scales with the video length $N$, and the average number of times that each value is seen during training scales as $K^2/N$. For long videos and small $K$, a look-up table will be both parameter-intensive and receive a sparse learning signal. We propose to parameterize $\mathbf{p}_{ij}$ with a learned function as $\mathbf{p}_{ij} := f_{\text{RPE}}(\mathbf{d}_{ij})$ where $\mathbf{d}_{ij} := pos(i) - pos(j)$. As shown in Fig. 7, $f_{\text{RPE}}$ passes a 3-dimensional embedding of $\mathbf{d}_{ij}$ through a neural network which outputs the vectors making up $\mathbf{p}_{ij}$. We use a network with a single $C$-channel hidden layer and were not able to measure any difference in the runtime between a DDPM with this network and a DDPM with a look-up table. Figure 8 shows the effect of the RPE network on attention

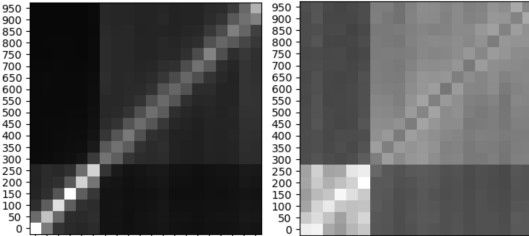

Figure 8: Temporal attention weights averaged over attention heads, spatial locations, network layers, and diffusion timesteps 1000 to 751. The color of entry $r, c$ is the average weight with which the $r$th frame in $\mathbf{x} \oplus \mathbf{y}$ attends to the $c$th frame. Black means zero weight. We plot an example where $\mathcal{Y} = \{0, 50, \ldots, 250\}$ and $\mathcal{X} = \{300, 350, \ldots, 950\}$. These plots are made with a network 50 000 iterations through training on the CARLA Town01 dataset. **Left:** From an architecture with an RPE network. **Right:** From a network with look-up tables of relative position embeddings. The architecture with an RPE network in the left plot has already learned to assign much greater weight to the nearest frames, while e.g. latent frames attend almost uniformly to other latent frames in the right plot. After training to convergence, both plots look similar to that on the left.

weights early in training. Architectures with an RPE network can learn the relative importance of other frames much more quickly. After training to convergence, there was no noticeable difference in sample quality but the architecture with an RPE network used 9.8 million fewer parameters by avoiding storing large look-up tables.

An alternative approach to our RPE network described by Wu et al. [38] shares the look-up table entries among "buckets" of similar $pos(i) - pos(j)$, but this imposes additional hyperparameters as well as restricting network expressivity.

## C  Explanation of our training task distribution

Here we provide our motivation, design choices and more explanation of our training task distribution, as visualized in Fig. 4 and implemented in Algorithm 2. Since we train our model to work with any custom sampling scheme at test time, our training distribution should be broad enough to assign some probability to any feasible choices of frames to sample and observe. At the same time, we want to avoid purely random sampling of frame positions (as in e.g. the ablation in Appendix A.2) as this will impair performance in realistic sampling schemes. Taking these considerations in mind, our design considerations for Algorithm 2 are simple:

1. The model should sample frames at multiple timescales, so we sample the spacing between frames (as on line 4 of Algorithm 2). A log-uniform distribution is a natural fit since events in a video sequence can happen over timescales in, e.g., seconds, minutes, or hours, and the differences between these are best captured by a log scale. The parameters of this log-uniform distribution are chosen to be the broadest possible (given the video length and the frame rate).

2. The user may wish to jointly sample multiple disparate sections of a video. We therefore make it possible to sample multiple groups of frames, potentially with different timescales (this is the purpose of the while loop in Algorithm 2).

3. The number of frames a user may wish to sample at a time is not fixed, so we add a broad uniform distribution over this (line 3 of the algorithm).

4. We train the model to perform conditional generation, so we choose groups of frames to be conditioned on (line 6 of the algorithm) using the simplest appropriate distribution, Bernoulli(0.5).

The remainder of the algorithm is boilerplate, gathering the indexed frames (line 1, 7, 9-13), randomizing the position of frames within the video (line 5) and enforcing that the number of frames does not exceed $K$ (line 8). Note that we do not claim that e.g. this exact mechanism for ensuring that $\leq K$ frames are sampled is a necessary or optimal choice for achieving FDM's performance. It is simply a design choice.

# D  Sampling schemes

Figure 9 illustrates each of the sampling schemes for which we reported results. We show the versions adapted to completing 300-frame GQN-Mazes videos from 36 initial observations, but all can be extended to e.g. different video lengths.

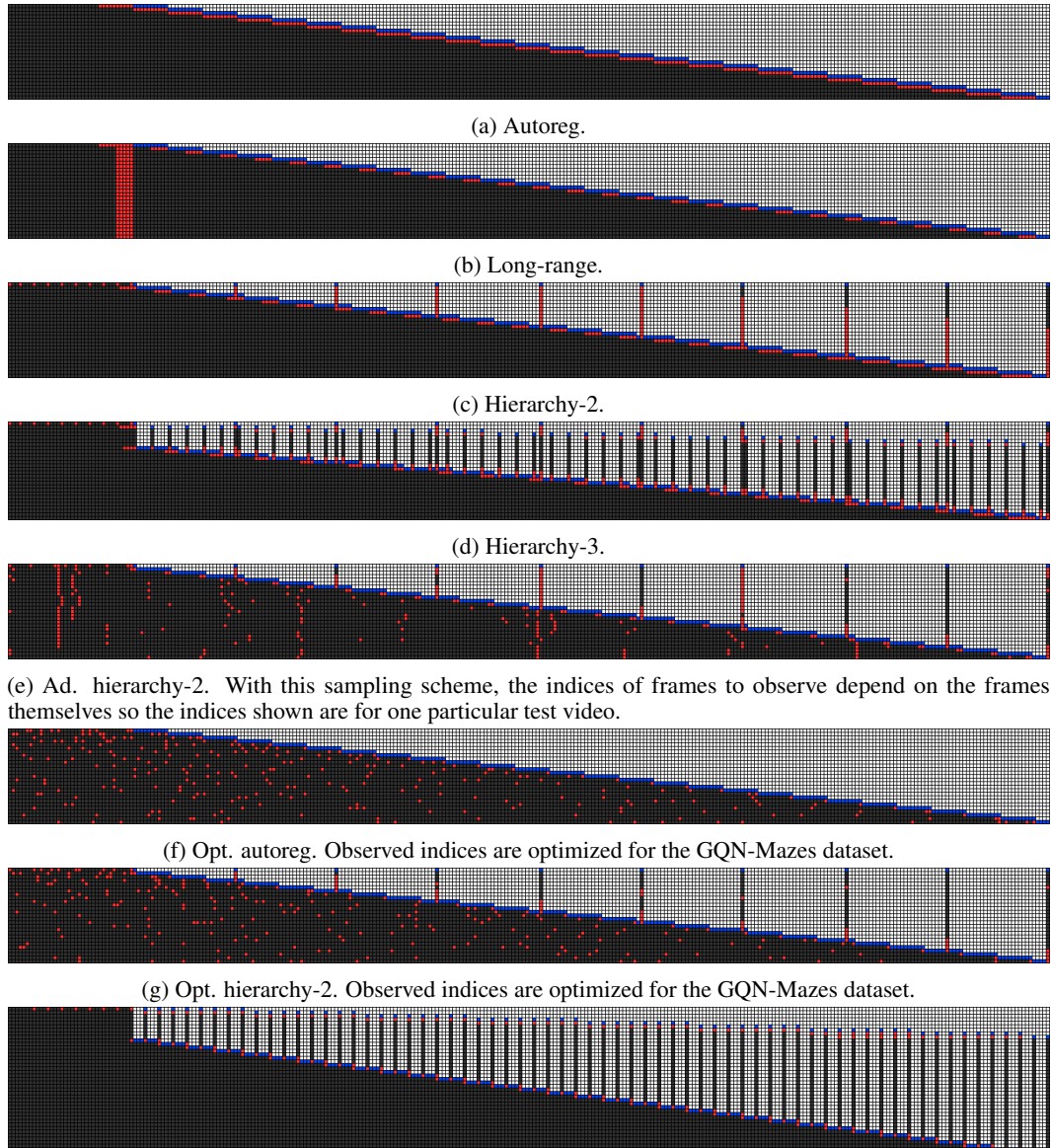

(a) Autoreg.

(b) Long-range.

(c) Hierarchy-2.

(d) Hierarchy-3.

(e) Ad. hierarchy-2. With this sampling scheme, the indices of frames to observe depend on the frames themselves so the indices shown are for one particular test video.

(f) Opt. autoreg. Observed indices are optimized for the GQN-Mazes dataset.

(g) Opt. hierarchy-2. Observed indices are optimized for the GQN-Mazes dataset.

(h) VDM. Like Ho et al. [16], we use two DDPMs to sample from this scheme.

Figure 9: Different sampling schemes used in experiments for the GQN-Mazes dataset.

## D.1  Adaptive sampling schemes

As mentioned in the main text, our Ad. hierarchy-2 sampling scheme chooses which frames to condition on at test-time by selecting a diverse set of observed of previously generated frames. Our procedure to generate this set is as follows. For a given stage $s$ we define $\mathcal{X}_s$ to be the same as the latent frame indices at the corresponding stage of the standard Hierarchy-2 sampling scheme. We then initialize $\mathcal{Y}_s$ with the closest observed or previously generated frame before the first index in $\mathcal{X}_s$, after the last index in $\mathcal{X}_s$, and any observed or previously generated frames between the first and

last indices of $\mathcal{X}_s$. We add more observed indices to $\mathcal{Y}_s$ in an iterative procedure, greedily adding the observed or previously generated frame with the maximum LPIPS [41] distance to it's nearest neighbour in $\mathcal{Y}_s$. Frames are added one-at-a-time in this way until $\mathcal{Y}_s$ is the desired length (generally $K/2$, or 10 in our experiments). Despite using a convolutional neural network to compute the LPIPS distances, the computational cost of computing $\mathcal{Y}_s$ in our experiments with Ad. hierarchy-2 is small relative to the cost of drawing samples from the DDPM.

## D.2 Optimized sampling schemes

We now describe in detail our procedure for optimizing the choice of indices to condition on at each stage in a sampling scheme. Our procedure requires that the "latent" frames are pre-specified. Figures 9f and 9g show examples of the indices that our optimization scheme chooses to condition on for GQN-Mazes when the latent indices are set according to either our Autoreg or Hierarchy-2 sampling scheme. We emphasize that the (relatively computationally-expensive) optimization described in this section need only be performed once and then arbitrarily many videos can be sampled. This is in contrast to the adaptive sampling scheme described in Appendix D.1, in which the sets of indices to condition on are chosen afresh (with small computational cost) for each video as it is sampled. For each stage of the sampling scheme, we select the set of indices to condition on with a greedy sequential procedure.

**Intialization of** $\mathcal{Y}$    This procedure begins by initializing this set of indices, $\mathcal{Y}$. In general $\mathcal{Y}$ can be initialized as an empty set, but it can also be initialized with indices that the algorithm is "forced" to condition on. We initialize it to contain the closest observed/previously sampled indices before and after each latent index. In other words, we initialize it so that there is a red pixel between any blue and gray pixel in each row of Figs. 9f and 9g.

**Appending to** $\mathcal{Y}$    On each iteration of the procedure, we estimate the DDPM loss in Eq. (4) (with uniform weighting) for every possible next choice of index to condition on. That is, we compute the DDPM loss when conditioning on frames at indices $\mathcal{Y} \oplus [i]$ for every $i \in \{1, \ldots, N\} \setminus \mathcal{X} \setminus \mathcal{Y}$. We estimate the loss by iterating over timesteps $t \in \{100, 200, \ldots, 1000\}$ and, for each timestep, estimating the expectation over $\mathbf{x}_0$ with 10 different training images. We found that the iteration over a grid of timesteps, rather than random sampling, helped to reduce the variance in our loss estimates. We then select the index resulting in the lowest loss, append it to $\mathcal{Y}$, and repeat until $\mathcal{Y}$ is at the desired length. We repeat the entire procedure for every stage of the sampling scheme.

# E    CARLA Town01

The CARLA Town01 dataset was created by recording a simulated car driving programatically around the CARLA simulator's Town01 [9]. The car is driven so as to stay close to the speed limit of roughly 3m/s where possible, stopping at traffic lights. The simulations run for 10 000 frames and we split each into 10 1000-frame videos.[8] Within each simulation, the weather and other world state (e.g. state of the traffic lights) is sampled randomly. The car begins each simulation in a random position, and navigates to randomly selected waypoints around the town. As soon as it reaches one, another is randomly sampled so that it continues moving. We use a 120 degree field of view and render frames at $128 \times 128$ resolution. To perform our evaluations on this dataset, we trained a regressor to map from a frame (either from the dataset or from a video model) to the corresponding town coordinates. This is trained with $(x, y)$ coordinates extracted from the simulator corresponding to the car location at each frame. The regressor takes the form of two separate networks: a classifier mapping each frame to a cell within a $10 \times 10$ grid placed over the town; and a multi-headed regressor mapping from the frame to $(x, y)$ coordinates in a continuous space. The final layer of the multi-headed regressor consists of 100 linear "heads", and which one to use for each data point is chosen depending on which cell the coordinate lies in. These two networks are trained separately but used jointly during evaluation, when the classifier is run first and its output determines which regressor head is used to obtain the final $(x, y)$ coordinate. We found that this approach improved the test mean-squared error considerably relative to using a single-headed regressor. The classifier was trained with data augmentation in the form of color jitter and a Gaussian blur, but we found that the multi-headed

---

[8]Due to technical glitches, not all simulations finished. When these occur, we simply save however many 1000-frame videos have been generated.

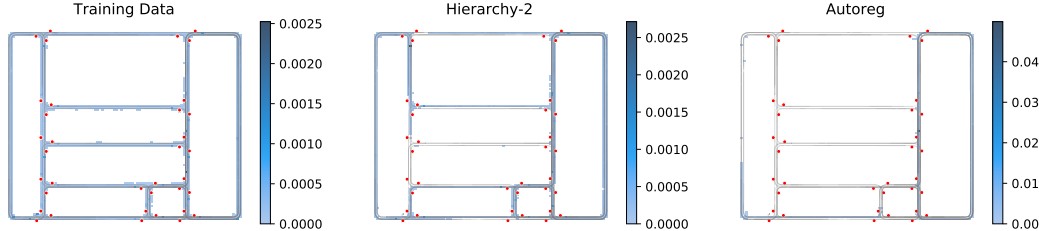

Figure 10: Heatmap of locations visited in (left) the CARLA Town01 training data, (middle) our long video sampled with Hierarchy-2, and (right) our long video sampled with Autoreg. The intensity of the blue color corresponds to the percentage of time spent in a given location and red dots mark the locations of traffic lights. Both sampled videos stop in locations in which the vehicle also stopped in the training data (shown as darker blue spots on the heatmap), corresponding to traffic light positions. The training data contains several days of video, so covers the map well. Each sampled trajectory lasts for 30-40 minutes so should not be expected to explore the entire map. However, Hierarchy-2 in particular obtains high coverage. The video sampled with Autoreg explores less of the map due to its tendency to remain stationary for long periods.

regressor did not benefit from this data augmentation so trained it without. Both the classifier and multi-headed regressor had the Resnet128 [14] architecture, with weights pretrained on ImageNet, available for download from the PyTorch torchvison package [23]. We will release the classifier and multi-headed regressor used to evaluate our models, enabling future comparisons.

## F    Sampled videos

To fully appreciate our results, we invite the reader to view a collection of FDM's samples in mp4 format.[9]  These include video completions, unconditionally sampled videos, and the long video samples from which some frames are shown in Fig. 1. To summarize the long video samples in this document, we visualize the trajectories taken throughout their (30-40 minute) course on CARLA Town01 in Fig. 10. Additionally, in the following pages, we show frames from uncurated samples of both video completions and unconditional video generations on each dataset.

---

[9]`https://www.cs.ubc.ca/~wsgh/fdm`

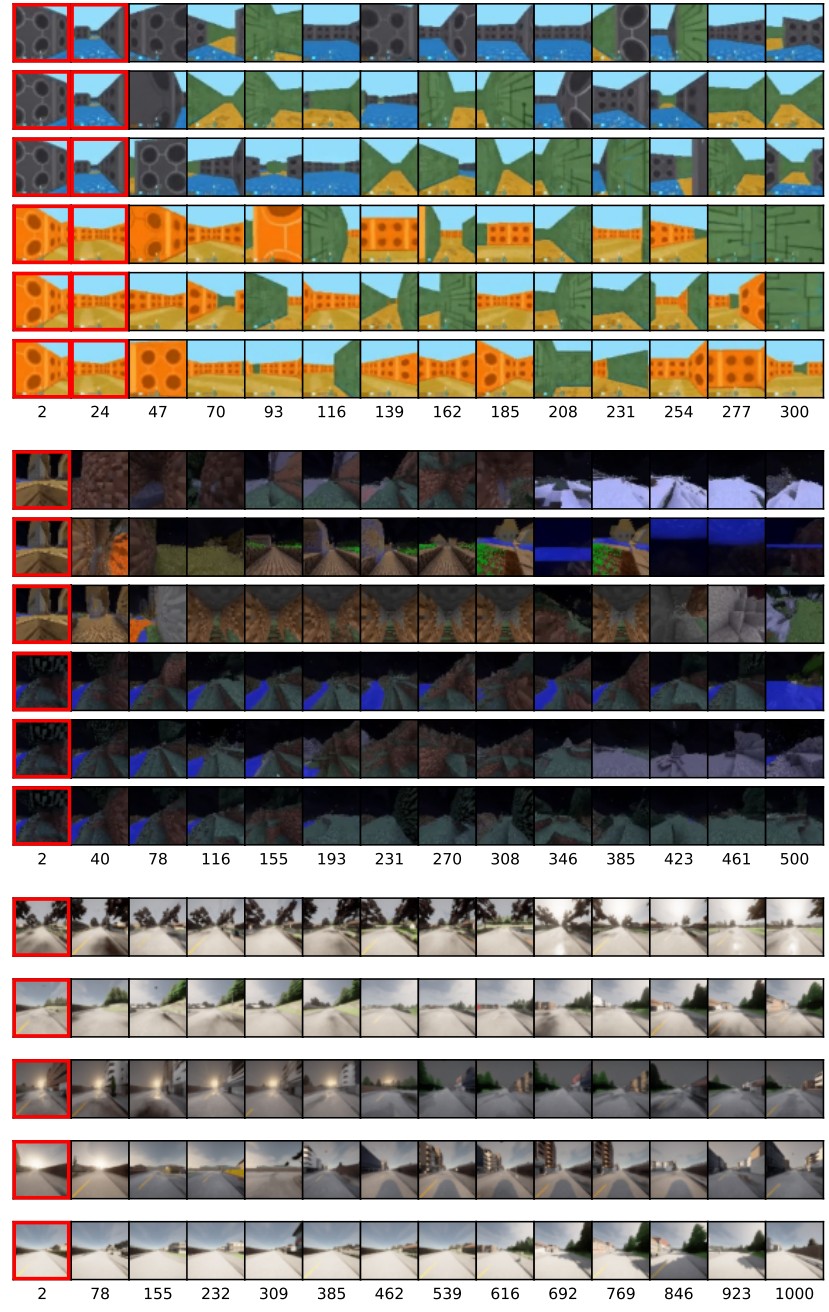

Figure 11: GQN-Mazes, MineRL, and CARLA Town01 completions sampled with Hierarchy-2. The first 36 frames are observed, indicated by a red border. Notably, the samples on GQN-Mazes do not exhibit the failure mode seen in Fig. 12.

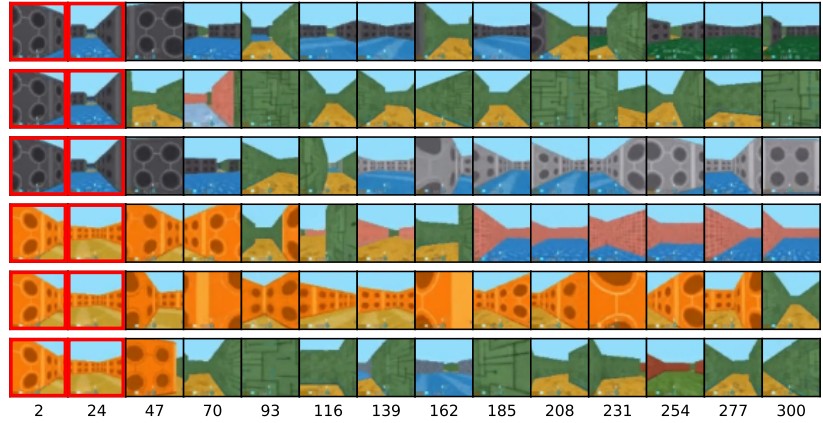

Figure 12: GQN-Mazes completions sampled with Autoreg. There should only be two wall/floor colors within each video, but Autoreg often samples more as it cannot track long-range dependencies. This issue is not seen in the Hierarchy-2 samples shown in Fig. 11.

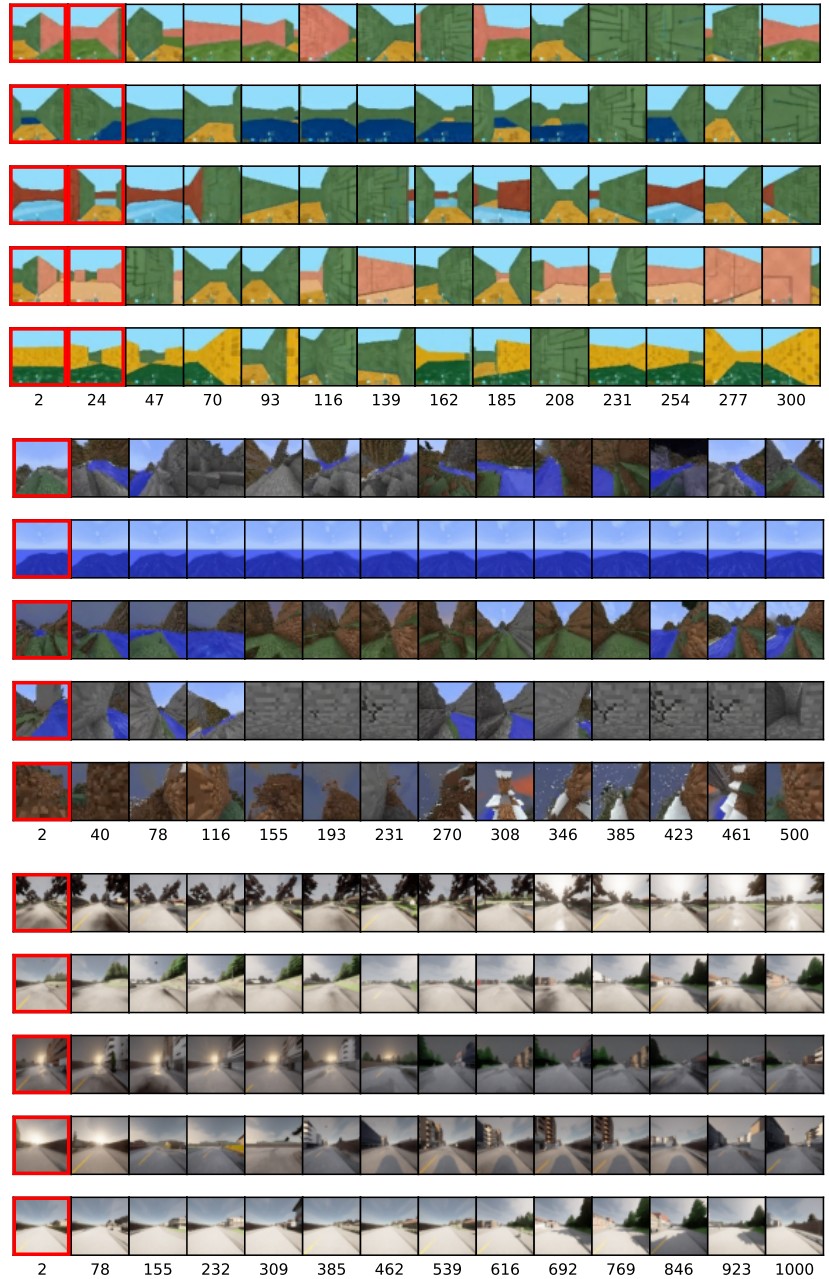

Figure 13: Videos sampled unconditionally by FDM with Hierarchy-2 on each dataset.