# OpenReview forum: "Flexible Diffusion Modeling of Long Videos"
_NeurIPS.cc/2022/Conference — NeurIPS 2022 Accept_

### Official Review · Reviewer_twbY · 2022-07-10

**Rating:** 4
**Confidence:** 5
**Soundness:** 2 fair
**Presentation:** 3 good
**Contribution:** 2 fair

**Summary:**

This paper proposes a video diffusion model that enables you to sample video by flexible conditioning(depending on your application scenarios). Technical side, the author inject the conditioning information by concatenating it(conditional images) with original images, and feed them into UNet to predict the score/noise of DDPM. The author designs a sampling strategy to enable the pretrained network can be  competent of flexible sampling.

For experiments, it mainly compares with Video Diffusion Model(VDM), CWVAE, in video completion task of three datasets(one of them is proposed by themselves). They also did several ablation studies to provide more insight about sampling schemes when testing, and sampling schemes when training.



**Questions:**

- More details about the comparison?
- More explanation about the "Con" parts of above session?

**Limitations:**

Yes

**Strengths And Weaknesses:**

Pro:

- The paper focuses on the video generation problem by diffusion method, which is a suitable, promising solution  by injecting the inductive-bias of grid-structured data like image, video, audio,etc.
- Clear illustration of the proposed idea about sampling strategy, training network, etc.
- The intuition of the method is technically sound in aiming to solve the complex sampling problem in diffusion models.

Con:

- Too few baselines to compare with, the author only compare with VDM a).in different datasets with them (they use UCF, which is way more complex than your dataset choice), b), they didn't release the code, lot's novelty impl of VDM is not included yet, like gradient-method, replacement method(check details from VDM), and also too few details are revealed to make the comparison with VDM not direct enough.
- Why In table 1, the gap of  loss between VDM and FDM in dataset CARLA Town01 is  so large? Typically, loss is not a good metric here to compare with your method and baselines in Table 1,2. As the model, the hyperparameter, the data preprocessing, and many details can deviate your conclusion from loss comparison.
- The idea about sampling training task is too simple, there is a lot of human bias  design inside, so I am not convinced by the proposed method especially when the baselines are too few.

---

> ### Author Response · Authors · 2022-08-02
> **Response to Reviewer twbY (1/2)**
>
> Thanks for your helpful comments. We’ll address your criticisms and describe the changes we've made in response below:
>
> ### “the gap of loss between VDM and FDM in dataset CARLA Town01”
> Thank you for raising this issue. Since submission, we have trained VDM for longer and this gap has decreased (so that VDM outperforms FDM’s Autoreg sampling scheme on FVD still but performs worse than FDM’s Hierarchy-2 sampling scheme on all metrics). Previously for CARLA Town01 we were approximately matching VDM’s total training compute with that of FDM. Since VDM consists of two networks (with frameskips 1 and 4), we now train each VDM network using as much training compute as the entirety of FDM, so that the VDM baseline receives twice as much training compute in total. This matches our protocol for GQN-Mazes and MineRL, and the relative performances of methods on CARLA Town01 are now similar to those on GQN-Mazes. Apologies for the confusion.
>
> ### “loss is not a good metric”
> This is a fair point, and we have moved these losses to Table 4 in the appendix, replacing them with the more informative “Accuracy” metric on GQN-Mazes. To clarify, these losses were the average DDPM loss (Equation 4) over the test set, such that an appropriate choice of $\lambda(t)$ would yield the ELBO of the test videos under each model and sampling scheme (although we use $\lambda(t):=1$ to de-emphasise pixel-level detail). They therefore should be broadly comparable between DDPMs with the same diffusion process hyperparameters. However, since the losses are difficult to interpret, possibly dominated by pixel-level detail, and can not be computed for CWVAE or TATS, we agree with the critique and have moved them to the appendix.
>
> ### “comparison with VDM not direct enough”
> Our VDM baseline can perhaps better be understood as an alteration of FDM in which we replace our primary contribution (meta-training objective compatible with arbitrary sampling schemes) with VDM’s sampling scheme and strategy of training two networks independently. As you say, due to VDM’s lack of a code release so far, there are aspects (gradient-method, architectural differences, etc.) in which this baseline differs from the VDM paper. However, as we argue in the overall response, these are orthogonal to our contributions and could be combined with them in future work. We therefore do not believe that showing improvements over the unreleased VDM model is as important as showing that our meta-learning objective presents an improvement over the VDM strategy of training both frameskip-1 and frameskip-4 models to use in a particular sampling scheme.
>
> ### “the baselines are too few”
> We are adding a comparison to TATS, as suggested by Reviewer wWqf. We hope this helps to provide additional context for our results. We also wish to reiterate here that, as we explained in our overall response, our experiments support all the claims of our paper, and provide strong reasons to believe that our contribution of a meta-training objective compatible with arbitrary sampling schemes can have high impact. Adding further baselines (such as TATS) is helpful for context, but the result of this comparison should not be what determines whether or not we have made a publishable contribution, especially when the amount of compute used by different research papers varies so considerably.
>
> ### “compare with VDM … in different datasets with them”
> We use different datasets to VDM because the setting we are interested in is the generation of long videos. On short videos (the UCF-101 videos modeled by VDM consist of only 16 frames), all frames can be sampled jointly and FDM reduces to be the same as our VDM baseline. While VDM is demonstrated on a text-conditional dataset with slightly longer (64 frame) videos, they did not release this dataset and it is anyway much shorter than the 300+ frame videos that we experiment with.

---

> ### Author Response · Authors · 2022-08-02
> **Response to Reviewer twbY (2/2)**
>
> ### “A lot of human bias design inside”
> Our aim in designing a training task distribution was to reduce the human bias inherent in many video modeling approaches, by allowing the sampling scheme to be tuned on data (whether through manual exploration or with our optimization scheme) instead of being set prior to training.
>
> We recognise, though, that the training task distribution in Algorithm 2 may seem to be imbued with human bias. After your comment, we have expanded our ablation analysis on the training task distribution in order to justify its structure. This experiment compares FDM against an ablation where the number of frames to use during training, the proportion of these to condition on, and the index of each frame are all sampled from a uniform distribution. We train networks on MineRL and GQN-Mazes, and draw samples from each with each of the sampling schemes in Table 1. We report FVDs below showing that the more structured distribution of FDM leads to improved performance with all tested sampling schemes.
>
> |           |                      | FDM | Uniform |
> |-----------|----------------------|-----|---------|
> |           | Autoreg              | **245** | 327     |
> |           | Hierarchy-2          | **235** | 279     |
> | GQN-Mazes | Long-range           | **198** | 281     |
> |           | Hierarchy-3          | **176** | 284     |
> |           | Adaptive hierarchy-2 | **178** | 281     |
> |           | Average                 | **226** | 296     |
> | | | | |
> |           | Autoreg              | **465** | 672     |
> |           | Hierarchy-2          | **586** | 902     |
> | MineRL    | Long-range           | **504** | 783     |
> |           | Hierarchy-3          | **515** | 970     |
> |           | Adaptive hierarchy-2 | **613** | 990     |
> |           | Average.                 | **518** | 786     |
>
> We also recognise that, as we try to be fully precise in Algorithm 2, it may appear excessively complex and mask the simplicity of the design choices behind it. We have therefore added the following explanation of our simple design choices to Appendix C:
>
> >Since we train our model to work with any custom sampling scheme at test time, our training distribution should be broad enough to assign some probability to any feasible choices of frames to sample and observe. At the same time, we want to avoid purely random sampling of frame positions (as in e.g. the ablation in Appendix A.2) as this will impair performance in realistic sampling schemes. Taking these considerations in mind, our design considerations for Algorithm 2 are simple:
> > - The model should sample frames at multiple timescales, so we sample the spacing between frames (as on line 4 of Algorithm 2). A log-uniform distribution is a natural fit since events in a video sequence can happen over timescales in, e.g., seconds, minutes, or hours, and the differences between these are best captured by a log scale. The parameters of this log-uniform distribution are chosen to be the broadest possible (given the video length and the frame rate).
> > - The user may wish to jointly sample multiple disparate sections of a video. We therefore make it possible to sample multiple groups of frames, potentially with different timescales (this is the purpose of the while loop in the algorithm).
> > - The number of frames a user may wish to sample at a time is not fixed, so we add a broad uniform distribution over this (line 3 of the algorithm).
> > - We train the model to perform conditional generation, so we choose groups of frames to be conditioned on (line 6 of the algorithm) using the simplest appropriate distribution, Bernoulli(0.5).
> >
> >The remainder of the algorithm is boilerplate, gathering the indexed frames (lines 1, 7, 9-13), randomizing the position of frames within the video (line 5) and enforcing that the number of frames does not exceed $K$ (line 8). Note that we do not claim that e.g. this exact mechanism for ensuring that $\leq K$ frames are sampled is a necessary or optimal choice for achieving FDM’s performance. It is simply a design choice.
>
> We hope that these changes have helped to address your concerns. If you have any that remain, please let us know and we will do what we can to address them. If not, please consider increasing your score.

---

### Official Review · Reviewer_wWqf · 2022-07-10

**Rating:** 7
**Confidence:** 5
**Soundness:** 3 good
**Presentation:** 4 excellent
**Contribution:** 3 good

**Summary:**

The paper presents a framework for generating long (many minutes, to hours) videos using diffusion models, extending to a series of progressively more complex environments such as Deepmind Lab, Minecraft, and CARLA (a self-driving simulator). As a key contribution, the authors introduce an efficient diffusion model that is trained to generate future frames based on a sparse selection of past frames. The authors investigate a series of selection methods, including various local or strided schedules, as well as a novel learned schedule.


**Questions:**

(1) Regarding my concern about learning long-range dependencies, it might provide a little more insight into the model if the authors were able to evaluate their GQN-Mazes model on the accuracy metric introduced in CW-VAE (https://proceedings.neurips.cc/paper/2021/file/f490d0af974fedf90cb0f1edce8e3dd5-Paper.pdf ). Note that only the Neurips version of the paper mentions it (arxiv was not updated). The relevant accuracy code can be found here: https://gist.github.com/vaibhavsaxena11/97b2d0a195c08ab2ed75cebb7d763799 .

(2) As mentioned before, it’s a little unclear to me what the purpose of the CARLA dataset is. Since it contains only videos of a small town, it should be fairly easy for a model to memorize the town’s layout. If we're aiming to test a model's ability to memorize a scene, I don't think that's particularly difficult or interesting.

(3) I think it would be useful to include some baselines on autoregressive models, such as TATS [1], or HARP [2], which can generate arbitrarily long videos using a sliding window with limited context




[1] Ge, Songwei, et al. "Long video generation with time-agnostic vqgan and time-sensitive transformer." arXiv preprint arXiv:2204.03638 (2022).

[2] Seo, Younggyo, et al. "Autoregressive Latent Video Prediction with High-Fidelity Image Generator." (2021).



**Limitations:**

Overall, the authors adequately addressed limitation and potential negative societal impacts of their work.

- As mentioned in the paper, diffusion models are slow to sample from, especially in the case of long-horizon video generation in which you may need to generate thousands or tens of thousands of frames.
- Sampling schemes, even when optimized, are dataset specific and not video specific, so the model is not able to dynamically retrieve context-dependent frames for generation


**Strengths And Weaknesses:**

Strengths
- The paper is well written and clear
- The authors present a novel diffusion model for videos that is simple and easy to scale.
- Experiments show high quality, visually consistent results on domains of varying difficulty, from more synthetic environments like DMLab to more realistic environments like CARLA

Weaknesses
- Novelty is a bit limited on the diffusion side, since it is a fairly direct application of image diffusion models to video

- I’m not too convinced at the ability for the model to accurately learn long-range dependencies, primarily since:

(1) FVD on GQN-Mazes is fairly sensitive to noise and will probably not capture long-range information such as maze layout.

(2) MineRL doesn’t really require long-range information since, as mentioned in the paper, it mainly consists of agents moving forward,
so only a few past frames are really needed for accurate video generation

(3) CARLA is a pretty small dataset, and it seems like a sufficiently large model can just memorize the town layout

---

> ### Author Response · Authors · 2022-08-02
> **Response to Reviewer wWqf**
>
> Thanks for your insightful comments. We’ll address your criticisms in turn:
>
> ### accuracy metric introduced in CW-VAE
> Thank you for this suggestion - we have added this accuracy metric to Table 1. Best performance is obtained by FDM with the Hierarchy-3 sampling scheme, although this is within the margin of error of the other hierarchical sampling schemes and the CWVAE. The autoregressive sampling schemes and VDM achieve considerably worse accuracies.
>
> ### it’s a little unclear to me what the purpose of the CARLA dataset is
> Our reason for training the model to memorize a single town’s layout is that it lets us use more semantically meaningful metrics for evaluation. While all methods tested can memorize the town layout to some extent, these metrics allow us to capture failure modes such as “jumping” unrealistically (via our OP metric), or becoming “stuck” at traffic lights (see Figure 6 and our WD metric). These metrics illuminate the considerable variation in the performance and failure modes of different methods (and indeed different FDM sampling schemes). We hope that being able to better understand and measure different types of failure can aid future research in video modeling.
>
> We agree that generalization to unknown scenes is a desirable property of a video model. While this is not assessed by the CARLA Town01 dataset, it is assessed in our other two datasets.
>
> ### I think it would be useful to include some baselines on autoregressive models, such as TATS
>
> We agree that more baselines would add helpful context. We are currently running the TATS method and hope to include results in the next few days.
>
> ### Sampling schemes, even when optimized, are dataset specific and not video specific
>
> While this is true for most of our sampling schemes, our Adaptive Hierarchy-2 sampling scheme does retrieve frames in a video-specific way (by choosing a set of frames to condition on which, roughly-speaking, are maximally different from one another). This is a very rough heuristic for selecting informative frames on-the-fly, but nonetheless produces our best FVD and WD scores on the CARLA Town01 dataset. We think that coming up with more principled ways to retrieve context-dependent frames (such as training a model to select the most informative frames) is an interesting avenue for future research.

---

> > ### Comment · Reviewer_wWqf · 2022-08-07
> > **Response**
> >
> > Thank you for your response. Given that the focus of the paper is on a flexible conditioning scheme (and not proposing a SOTA video generation model), then I believe that it would be important for the authors to motivate certain aspects of their design decisions through further explanations or ablations, specifically:
> >
> > 1) Why is the distributions of latent and conditional frames chosen the way it is (positions sampled uniformly, stride sampled log-uniformly)? How does this compare to other sampling distributions (e.g. naively uniform)? How much does this matter?
> >
> > 2) Is there a train-test time gap in performance? For example, if you trained a separate FDMs constrained to each sampling scheme (AR, long-range, hierarchical, etc.), is there a large performance gap compared when training using random sampling / conditioning schemes?

---

> > > ### Author Response · Authors · 2022-08-07
> > > **Re: Response**
> > >
> > > Thanks for the extra comments!
> > >
> > > ### Question 1
> > > In the most recent revision, we added an explanation of these design decisions to Appendix C and to the end of [Response to Reviewer twbY (2/2)](https://openreview.net/forum?id=0RTJcuvHtIu&noteId=nMorZVnbWTY) (motivated by Reviewer twbY’s comments). We also have an ablation comparing networks trained using this distribution to networks trained through uniform sampling of the frame positions. On both GQN-Mazes and MineRL, and with every sampling scheme tested, networks trained using our structured task distribution outperformed those trained using uniform sampling. See Appendix A.2 for full details, and the table of FVD scores (lower is better) for this ablation in [Response to Reviewer twbY (2/2)](https://openreview.net/forum?id=0RTJcuvHtIu&noteId=nMorZVnbWTY).
> > >
> > > ### Question 2
> > > There is a train-test gap. We investigated this for the autoregressive sampling scheme in the paragraph starting on line 271. Specializing training to the autoregressive sampling scheme instead of using FDM led to a 5.1% and 16.7% improvement in FVD on GQN-Mazes and MineRL respectively.
> > >
> > > To re-emphasise FDM’s advantages over such a fixed training task, we point out that FDM is still beneficial since the optimal sampling scheme is not generally known a priori. Using FDM allows both manual exploration of the space of sampling schemes, and automated optimization of the sampling scheme (as described in the paragraph starting on line 182). For a computational budget large enough to train multiple models, best results may be obtained by using FDM to explore the space of sampling schemes and then training a second diffusion model specialized to a single optimized sampling scheme. For users who wish to train only a single model, we would still recommend using FDM rather than e.g. just training a model compatible with Autoreg. This is because the performance of each sampling scheme varies greatly between datasets and so e.g. although in Table 1 Autoreg is the best sampling scheme on MineRL, it is the worst on GQN-Mazes and CARLA Town01.
> > >
> > > Ideally, in addition to these results for specializing training to Autoreg, we would respond with results for specializing training to the other four sampling schemes in Table 1. Unfortunately, the author-reviewer discussion period ends on Tuesday and there is not enough time to obtain results before then. Nonetheless, we will include these results if accepted.

---

### Official Review · Reviewer_sEEi · 2022-07-12

**Rating:** 4
**Confidence:** 4
**Soundness:** 3 good
**Presentation:** 2 fair
**Contribution:** 2 fair

**Summary:**

The paper proposes a DDPM based model for video frame prediction. It uses a randomized masking the randomly condition some frames on others allowing for long term predictions. Authors tested different sampling strategies on a newly introduced dataset to demonstrate its capability in generating long videos. The paper compares its result to CWVAE and Video Diffusion Models.

**Questions:**

Please refer to weaknesses for my questions. Here is a short list:
- Is the model overfitting or can it generalize to unseed cases?
- How does the model perform vs a simpler DDPM method.
- How does the model compare vs other models is other datasets (CWVAE Table 1 from arxiv has a good list)

**Limitations:**

Yes

**Strengths And Weaknesses:**

===== Strengths:
- The randomized masking algorithm is simple and effective particularly given the capability of limiting it to a fixed number of frames to put an upper bound on computation.
- The paper is well written, easy to follow and read.
- The new dataset is effective in demonstrating the capability of the proposed method for predicting long videos in a dynamic setting.

==== Weaknesses:
- Analysis. There are multiple points that the paper can be improved by adding more analysis. For example, while the proposed masking strategy makes sense, there is no experiment to show how effective it is vs simpler strategies such as just masking every 10 frames. I understand that the proposed method allows for different sampling strategy but it would be great to show how a simple autoregressive + infilling sampling with a simple masking strategy perform on a similar problem.

- Comparisons. The paper only compares its results to other models in Table 1 which is not enough for a fair comparison. I suggest authors to include more comparisons. CWVAE has more dataset numbers, metrics and baseline numbers to compare to. FitVid also has a section on long video predictions (with overfitting) on Human3.6M. I also recommend comparing the model with other video prediction models on shorter videos, given the richer number of models in that domain which can help demonstrating the capability of DDPMs for video prediction.

- Overfitting. Another possible improvement is to condition the model on an out of distribution data e.g. an unseen street. Given the size of the dataset, my main concern is that the model memorized everything and is not capable of generalization which is the main goal. FitVid did similar analysis and found overfitting as the main reason for their capability of generating super long videos.

- Qualitative results. Given the nature of videos, I highly encourage authors to include videos on an anonymous website for better demonstration and comparison.

---

> ### Author Response · Authors · 2022-08-02
> **Response to Reviewer sEEi (1/3)**
>
> Thanks for your insightful comments. We’ll address your criticisms below and describe the changes we've made in response:
>
> ### “I highly encourage authors to include videos on an anonymous website”
>
> Thank you for this suggestion. We now anonymously host samples from FDM and our baselines at https://fdmolv.github.io. These results make clear our improvements over CWVAE and VDM and also the importance of the choice of FDM sampling scheme. We had previously included a link to an anonymous google drive in the appendix, but we apologize for a (now-fixed) issue with the pdf that meant it was not clickable or searchable. We now link to the website from the manuscript.
>
> ### “while the proposed masking strategy makes sense, there is no experiment to show how effective it is vs simpler strategies”
>
> Please correct us if we have misinterpreted, but we believe that by “masking strategy” you mean our training task distribution (as opposed to e.g. a test-time sampling scheme). We have therefore added further analysis of our training task distribution (to Appendix A and the last paragraph of Section 6). We do so by extending our comparison between FDM and an ablation of FDM with a “uniform” training task distribution. In the uniform task distribution the number of frames to observe, the proportion of these to condition on, and the index of each frame are all sampled from a uniform distribution.
> We train networks with each of these training task distributions on MineRL and GQN-Mazes, and draw samples from each with each of the sampling schemes in Table 1. We report FVDs below:
>
> |           |                      | FDM | Uniform |
> |-----------|----------------------|-----|---------|
> |           | Autoreg              | **245** | 327     |
> |           | Hierarchy-2          | **235** | 279     |
> | GQN-Mazes | Long-range           | **198** | 281     |
> |           | Hierarchy-3          | **176** | 284     |
> |           | Adaptive hierarchy-2 | **178** | 281     |
> |           | Average                 | **226** | 296     |
> | | | | |
> |           | Autoreg              | **465** | 672     |
> |           | Hierarchy-2          | **586** | 902     |
> | MineRL    | Long-range           | **504** | 783     |
> |           | Hierarchy-3          | **515** | 970     |
> |           | Adaptive hierarchy-2 | **613** | 990     |
> |           | Average.                 | **518** | 786     |
>
> We see that the FDM training task distribution is better than either alternative for almost all sampling scheme and dataset combinations. We have added these results to the appendix and refer to them in the final paragraph of Section 6. All runs in this table use the hyperparameters reported in the bottom two rows of Table 3 (explaining the disparity between these FVDs and those in Table 1).
>
> ### It would be great to show how a simple autoregressive + infilling sampling scheme with a simple training task distribution performs on a similar problem.
>
> This suggestion sounds similar to our VDM baseline, which uses an autoregressive + infilling sampling scheme. It uses extremely simple training task distributions since it consists of two networks, each of which is trained to model a fixed number of frames with fixed spacing in between.

---

> ### Author Response · Authors · 2022-08-02
> **Response to Reviewer sEEi (2/3)**
>
> ### “CWVAE has more … metrics”
>
> We have computed the SSIM, PSNR and LPIPS metrics used by CWVAE, as well as the accuracy metric on GQN-Mazes suggested by Reviewer wWqf. We report them below, as well as in Tables 1 and 4.
>
> |       |                 |                    | GQN-Mazes |          |          | MineRL   |          |          | CARLA Town01 |          |          |
> |-------|-----------------|--------------------|-----------|----------|----------|----------|----------|----------|--------------|----------|----------|
> |       |                 | Mazes accuracy (%) | LPIPS     | SSIM     | PSNR     | LPIPS    | SSIM     | PSNR     | LPIPS        | SSIM     | PSNR     |
> | CWVAE | CWVAE           | 82.6               | 0.41      | **0.64** | 16.3     | 0.50     | **0.59** | **19.3** | 0.53         | 0.71     | 15.5     |
> | VDM   | VDM             | 77.8               | 0.39      | 0.61     | 16.1     | 0.33     | 0.54     | 19.2     | 0.35         | 0.71     | 15.4     |
> |       | Autoreg         | 69.6               | 0.40      | 0.60     | 15.5     | **0.32** | 0.53     | 18.9     | 0.28         | 0.74     | 17.5     |
> |       | Long-range      | 77.0               | **0.37**  | 0.61     | 16.3     | **0.32** | 0.54     | 19.0     | **0.26**     | **0.75** | **18.5** |
> | FDM   | Hierarchy-2     | 82.8               | **0.37**  | 0.61     | **16.4** | 0.33     | 0.54     | 19.0     | 0.29         | 0.73     | 17.2     |
> |       | Hierarchy-3     | **83.8**           | 0.38      | 0.62     | **16.4** | 0.33     | 0.54     | 19.1     | 0.31         | 0.72     | 16.9     |
> |       | Ad. hierarchy-2 | 83.2               | **0.37**  | 0.62     | **16.4** | 0.33     | 0.53     | 19.0     | 0.30         | 0.72     | 17.0     |
>
> The GQN-Mazes accuracy metric is computed by classifying videos into one of three categories and then computing what proportion of video completions match the category of the ground truth. This helps to assess which models are accurately capturing long-range dependencies. Best performance is obtained by FDM with the Hierarchy-3 sampling scheme, although this is only slightly higher than that of the other hierarchical sampling schemes and the CWVAE. The autoregressive sampling schemes and VDM do considerably worse.
>
> Following standard procedure (see e.g. FitVid), the SSIM, PSNR, and LPIPS metrics reported are for the best of k sampled videos on each test video (where k is 5 for GQN-Mazes, 5 for MineRL, and 1 for CARLA Town01). Under the LPIPS metric, the relative ordering of the methods is roughly consistent with that under the previously-reported FVD scores and the visual quality (see https://fdmolv.github.io/mazes, https://fdmolv.github.io/minerl, https://fdmolv.github.io/carla), although differences between the methods are smaller. Under the SSIM and PSNR metrics, the relative ordering is very different, and the CWVAE often achieves the best performance. This is extremely counter-intuitive given the blurry samples produced by CWVAE (see https://fdmolv.github.io/minerl in particular). We attribute this to the PSNR, SSIM, and LPIPS metrics’ known issues for evaluating stochastic tasks, in which they favor blurry predictions. This is especially troublesome for SSIM and PSNR, which are closely related to mean-squared error in pixel space. For this reason, we place the SSIM, PSNR and LPIPS results in the appendix along with some discussion.
>
>
>
> ### Other baselines
>
> We agree that using more baselines would help to add context to our experiments and so, as mentioned in our overall response, are currently evaluating the TATS method (https://arxiv.org/abs/2204.03638) suggested by Reviewer wWqf and hope to post results during the discussion period.
>
> Regarding the CWVAE’s baselines, we note that they do not generally outperform the CWVAE in its experiments and produce blurry samples (see e.g. in Figure F.1 of https://arxiv.org/pdf/2102.09532.pdf). We show, in both the qualitative results at https://fdmolv.github.io/ and in Table 1, that diffusion-based video models considerably outperform CWVAE. We therefore do not think that these baselines are likely to be competitive with FDM and have chosen to prioritize evaluating the TATS baseline mentioned by Reviewer wWqf in this rebuttal period. We also appreciate the suggestion of FitVid, but note that its training time (12 days on 16 TPUs) is prohibitive for us, especially in this short rebuttal period.

---

> ### Author Response · Authors · 2022-08-02
> **Response to Reviewer sEEi (3/3)**
>
> ### Shorter videos
>
> While we appreciate the suggestion of experimenting on shorter videos (and your suggestion about testing extrapolation from short training videos to long test samples like FitVid), we note that for training on small video lengths, FDM reduces to be almost the same as our VDM baseline. Given that the VDM paper trains models with orders of magnitude more computation than is available to us, we do not expect to be able to produce results as good as those in the VDM paper, and so leave the VDM paper to serve as a demonstration of the use of diffusion model for short videos.
>
> ### Overfitting
>
> Regarding generalization on GQN-Mazes and MineRL (in which the world is procedurally generated afresh for each video), we have added videos to https://fdmolv.github.io/train-vs-test-completions comparing completions for training set videos with completions for test set videos. We note that (a) the quality of completions is visually similar in each case and (b) the completions are stochastic even on training set videos (in the sense that different samples are different from each other), giving indications that FDM achieves some level of generalization.
>
> Regarding the CARLA Town01 dataset, we wish to make clear that generalization beyond the data distribution is not our goal, or expected. We see generalization from training videos to held-out test videos from the same Carla town, but not to videos from other towns. As explained in Section 5, we expect that “videos sampled from [models trained on CARLA Town01] will be recognisable as corresponding to routes [within Town01]”. This is because the size of our training set is sufficient to traverse Town01 over 10 times. If our goal was to generalize to any town, this could be described as overfitting to one town. This is not, however, our goal. Our goal is to train models to generate coherent trajectories within Town01, and then use the semantically-meaningful OP and WD metrics to understand their failure modes. We have added sampled videos to https://fdmolv.github.io/carla-out-of-distribution, confirming that our model can generate coherent completions of held-out test videos inside Town01 but, as expected (and in common with our baselines), does not produce coherent completions when prompted to continue videos starting in other towns.
>
> We hope that our response and updates have helped to address your concerns. If any concerns remain, please let us know what they are and what may be done to address them. If not, please consider increasing your score.

---

### Author Response · Authors · 2022-08-02
**Overall response**

Thank you to all the reviewers for their time and helpful comments.

As Reviewer sEEi suggested, we have created an anonymous website (https://fdmolv.github.io/) containing sampled videos from FDM and our baselines. This is helpful for understanding the improvements brought about by our contributions, as well as various failure cases, and we would be grateful if the reviewers view the website at their convenience.

In response to the suggestions of additional baselines from several reviewers, we are adding a comparison to the concurrent TATS video model suggested by Reviewer wWqf. We would additionally like to reiterate our primary contribution and ensure we are in agreement with the reviewers with regard to the purpose of our baselines. Our primary contribution is the idea of training a single model to work with any custom sampling scheme chosen at test-time. We include the VDM baseline to demonstrate the benefits of this contribution over VDM’s less general approach that trains a model to work with a single sampling scheme. Our CWVAE and (in-progress) TATS baselines situate both our model and our VDM baseline with respect to alternative (non-diffusion based) approaches. We intentionally do _not_ claim to achieve state-of-the-art results on video generation benchmarks, or demonstrate the scalability of DDPM-based video models to extremely complex datasets. For such results see Google’s VDM, which runs most experiments on 128 TPUs. In contrast our models are trained on 1 to 4 GPUs. Nonetheless, our primary contribution is orthogonal to those of VDM and e.g. MCVD (https://arxiv.org/abs/2205.09853) and so could likely have a large impact if directly combined with them. For these reasons, and given the TATS baseline in progress and further ablations added (see individual responses) in response to the reviewers’ comments, we believe that our experiments both demonstrate the soundness of our contribution and show that it is of interest to the NeurIPS community.

We’ll now respond to each reviewer’s comments in turn.

---

### Author Response · Authors · 2022-08-09
**Update to our review response**

As earlier stated, we have been working on running the concurrent TATS method in our setting. We present samples on GQN-Mazes [here](https://fdmolv.github.io/tats). In these results, the VQGAN and the transformer are trained for 1-2 days each on 8 GPUs. This leads to a total cost roughly 3x that of our FDM model trained on the same dataset. Despite this, the results are worse than any of our baselines, deteriorating within a few frames into a shaky close-up of a wall. We suspect that this method may simply need much more training time than FDM to perform well. We commit to training TATS for longer if accepted and computing quantitative metrics, as well as performing more hyperparameter searches to optimize TATS for our lower computational budget. We hope that these, albeit incomplete, results help to provide more context for the good performance of FDM.

We thank Reviewer wWqf for their engagement. We believe that we have addressed the concerns raised by Reviewers twbY and sEEi. We would therefore appreciate it if they would either raise their scores or explain why they do not think this paper is fit for publication.

---

### Meta-Review · Area_Chair_aZYT · 2022-08-30

**Recommendation:** Accept
**Confidence:** Less certain

**Metareview:**

This paper proposes one of the first video diffusion models (concurrently with a couple other papers) and presents an architecture that admits many different approaches to sampling a long video. Video generation as important but unsolved problem, and this paper makes an important step towards long video generation. I do agree with reviewers that more baselines can be beneficial, and the authors have agreed to include at least one more baseline. Regardless, I believe the experiments in the paper add to the growing literature on diffusion models and will be interesting to the NeurIPS audience.

**Award:**

No

---

### Decision · Program_Chairs · 2022-09-14

Accept